# ß-adrenergic-like signalling engages *CrebB* in *Drosophila* gut to promote female longevity

Ahmed F. Sumit [1,2], Ben Whitehead[1], Ilgaz Özyeşil[1], Dunxin Shen[1], Shajahan Anver[1], Lazaros C. Foukas [1] & Nazif Alic [1] ✉

Adrenergic signalling is heavily implicated in human age-related disease and yet the potential for this neuroendocrine signalling pathway to modulate ageing has received little attention. Here, we use *Drosophila melanogaster* to test if adrenergic-like signalling can promote longevity by manipulating tyramine (TA) or octopamine (OA), the invertebrate equivalents of adrenergic hormones. Increased neuronal synthesis of TA boosts health and longevity in both sexes, whereas OA is marginally beneficial in males. Orally administered TA or OA extend female or male lifespan, respectively. Increased activation of the ß-adrenergic-like signalling in the gut, by manipulating a ß-adrenergic-like receptor, *PKA* or *CrebB*, is sufficient to delay female ageing. Transcriptional profiling reveals that *CrebB* links the ß-adrenergic pathway to longevity-promoting processes in the gut where its function is required for the beneficial effects of TA feeding in females. Here we show that localised activation of ß-adrenergic signalling has the potential to counter animal ageing.

Adrenalin and noradrenalin are biogenic amines produced by neuronal and endocrine cells to regulate vertebrate physiology. Initially appreciated as stress-induced signals that promote survival under acute stress by increasing alertness, energy mobilisation, heart rate, and altering blood flow in what has been termed the "fight or flight" response[1,2], they are now known to have a plethora of effects on numerous cell types, such as muscle, endothelial, epithelial, immune cells, and adipocytes[3–9]. These diverse effects underpin their ability to profoundly alter animal physiology.

Adrenergic signalling is implicated in a plethora of human age-related diseases and dysfunctions, from cardiovascular disease, obesity, and chronic obstructive pulmonary disease to memory impairments and urinary incontinence[7,10–13]. Adrenergic signalling has often been a successful target for disease therapy: ß-adrenergic antagonists have been used clinically for treatments of cardiovascular disease since the 1960s[14], while agonists have proven useful in chronic obstructive pulmonary disease and urinary incontinence[7,11]. However, despite the importance of adrenergic signalling in ensuring vertebrate homeostasis and survival, and despite its intimate involvement in numerous human age-related diseases and its pharmacological relevance, the direct role of this neuroendocrine pathway in promoting longevity has received surprisingly little attention.

Genetic tractability and a relatively short lifespan have facilitated the use of *Drosophila melanogaster* to demonstrate the evolutionary conservation of neuroendocrine pathways modulating animal ageing[15,16]. Insects, such as *Drosophila*, have a signalling system initiated by the biogenic amines tyramine (TA) and octopamine (OA), that performs similar functions and is evolutionarily related, albeit not orthologous, to the vertebrate adrenergic signalling[2,17]. Both TA and OA play important and versatile roles in adult *Drosophila* physiology, which include regulation of neuromuscular, metabolic, and reproductive organ systems[2,18–22]. Here, we examine whether increased activity of this neuroendocrine signalling pathway can promote health and survival in old age in *Drosophila*.

Using genetic tools to increase neuronal TA and OA production, and oral administration of the two ligands, we show that they have a sexually dimorphic effect on adult survival: OA has a minor, beneficial effect on male survival, while TA consistently promotes female longevity. We find that this beneficial effect of TA can be recapitulated by the activation of the ß-adrenergic-like signalling pathway in the female

[1]Institute of Healthy Ageing, Department of Genetics, Evolution and Environment, University College London, Gower St, London, UK. [2]Department of Genetic Engineering and Biotechnology, University of Dhaka, Science Complex Building, Dhaka University Campus, Dhaka, Bangladesh. ✉e-mail: n.alic@ucl.ac.uk

gut; indeed, TA requires the transcription factor *CrebB* in the gut to extend lifespan, where *CrebB* links ß-adrenergic-like signals to a longevity-promoting transcriptional programme. Our findings show that localised ß-adrenergic activation can delay ageing of the whole animal.

## Results

### Increased production of TA in tyraminergic neurons extends lifespan

We set out to test whether a manipulation of TA or OA signalling could extend lifespan. Experimental manipulations that result in longevity, rather than a shortened lifespan, reveal factors that limit wild-type survival into old age. We speculated that increased TA or OA could promote longevity for the following reasons. TA and OA levels decrease with age, with TA levels peaking during later stages of development[23], while OA levels are highest in young adults[23–25]. The levels of their precursor, tyrosine, also decline with age in the wild type, and preventing this decline can promote longevity[26]. Lastly, flies lacking OA are short-lived[20]. Hence, we opted to test the effect of increased TA or OA production on lifespan, focusing initially on TA.

TA is produced from tyrosine by the action of Tyrosine decarboxylase (Tdc)[2,21] (Fig. 1a). This enzymatic activity is encoded by two genes in *Drosophila* and one of these, *Tdc2*, has been extensively characterised. Its expression is restricted to discrete populations of neurons (tyraminergic neurons)[19]. We wanted to upregulate *Tdc2* expression specifically within its natural domain of expression. We used a driver generated previously where *GAL4* is expressed from the *Tdc2* promoter in a manner shown to recapitulate native *Tdc2* expression[19]. We combined this native promoter-*GAL4* fusion (*Tdc2-GAL4*) with *UAS-Tdc2*, in a healthy, outbred background. The resulting phenotypes were assessed in both sexes. To ascertain sexual dimorphism in our statistical analyses, we used regression models with sex as a covariate. To robustly identify the specific effects of overexpression, we used a priori contrasts to compare *Tdc2>Tdc2* flies, carrying both the driver and the transgene, to both *Tdc2-GAL4-* and *UAS-Tdc2-*alone controls, as well as the two controls to each other; the two controls may show phenotypic differences due to insertional mutagenesis, unintended effects of the constructs or genetic drift[27]. The results of statistical analyses are shown in Supplementary Figs.

Increased expression of *Tdc2* resulted in increased levels of TA present in adult females and males (Fig. 1b), with a stronger effect observed in females than males (Fig. 1b, Supplementary Fig. 1a). We found no detectable increases in OA (Fig. 1c, Supplementary Fig. 1b). These increased levels of TA were sufficient to extend both female and male lifespan, with over 20% extension of median lifespans and no significant sexual dimorphism in the longevity induced (Fig. 1d, Supplementary Fig. 1c). The effect was robust as a similar extension was observed in a completely independent experimental trial (Supplementary Fig. 1d and e; all independent lifespan repeats are presented in Supplementary Figs.).

The effects of neuronal TA were pleiotropic, as expected[2,21]. *Tdc2>Tdc2* flies were slightly smaller (Supplementary Fig. 1f and g), indicating altered development. Females laid fewer eggs (Supplementary Fig. 1h); neuronal TA may directly inhibit egg-laying as suggested previously[19]. Using both proboscis-extension and food-consumption assays, we observed that *Tdc2>Tdc2* females fed less than controls (Supplementary Fig. 1i), consistent with the role of TA in regulating feeding behaviour[22]; this reduced nutrient consumption may contribute to their longevity, however, reduced feeding was not observed in males (Supplementary Fig. 1j) who were also long-lived.

Extended lifespan, reduced body size and reduced egg laying are reminiscent of phenotypes resulting from decreased insulin-like signalling[15,16,28,29]. To test whether increased *Tdc2* expression dampens the activity of this signalling pathway, we examined the phosphorylation status of two kinases that are activated downstream of the

insulin receptor, AKT and ERK[30], focusing on females, since insulin-like signalling is better characterised in this sex. We found no evidence of a decreased phosphorylation of either AKT or ERK (Fig. 1e, f, Supplementary Fig. 1k). Rather, ERK phosphorylation was slightly increased, which is consistent with the suggestion that increased TA may cause a retention of eggs[19] in which EKR is known to be active[31]. Overall, we found that increased production of TA from tyraminergic neurons extended lifespan in both sexes, without an observable decrease in insulin-like signalling.

### Increased TA delays neuromuscular and intestinal ageing

We next examined the effect of increased TA production in tyraminergic neurons on indices of health in old age. Age-related decline in neuromuscular function can be observed as a reduced ability to climb a vertical surface in the negative geotaxis assay[32]. In young flies (day 7), both males and females with increased *Tdc2* expression climbed less well than the two controls (Fig. 2a, Supplementary Fig. 2a), possibly due the inhibitory effects of TA on locomotion[18]. However, the climbing ability of the controls declined rapidly with age, while the decline in *Tdc2>Tdc2* flies was slower so that by day 42 the flies with over-expressed *Tdc2* climbed better than the controls (Fig. 2a). We used a *linear model* (*LM*) to analyse these data. The analysis confirmed that the age-related decline in climbing ability was significantly reduced in *Tdc2>Tdc2* flies compared to both controls in both sexes (observed as significant genotype-by-age interaction, Supplementary Fig. 2a), indicating preservation of neuromuscular health with age.

The health of the fly digestive system also deteriorates with age, resulting in increased gut permeability[33] and hyperproliferation and mis-differentiation of intestinal stem cells (ISCs)[34]. We assessed the gut barrier function by observing the frequency of flies with leaky guts at different ages. The gut barrier function was also better maintained in old age in *Tdc2>Tdc2* flies compared to the two controls in both sexes (Fig. 2b). This was confirmed by statistical analysis using *ordinal logistic regression* (significant genotype-by-age interaction, Supplementary Fig. 2b). Furthermore, the age-related gut dysplasia, observed as number of cells marked with phosphorylated histone H3, was also reduced in females (Supplementary Fig. 2c). Overall and in addition to the extended lifespan, increased production of TA in tyraminergic neurons resulted in broad improvements in old-age health, affecting both the neuromuscular and digestive systems in both sexes.

### Neuronally produced OA modestly promotes male longevity

TA and OA have distinct physiological roles[2,21]. OA is produced from TA by the action of TA β-hydroxylase (Tbh)[2,21] (Fig. 3a). The sole gene encoding this activity, *Tbh*, is also expressed in a specific population of neurons (octopaminergic neurons) and not in any non-neuronal tissues[35]. To test the role of OA in longevity, we increased expression of *Tbh* using *UAS-Tbh* and a *GAL4* driver that, like the *Tdc2-GAL4* used above, was generated using an enhancer fragment from the *Tbh* gene[36]. Note that the expression of this driver has been confirmed to populations of neurons[36] but it is unlikely to recapitulate the full expression pattern of the *Tbh* gene as different *Tbh* enhancer appear to be required for expression in different octopaminergic neurons[36,37]. Driving *Tbh* with this driver did not result in significant increases in OA levels in whole-fly extracts, even though OA levels were somewhat higher in *Tbh>Tbh* females than the controls (Fig. 3b, Supplementary Fig. 3a), possibly due to the limited *Tbh* expression domain, or fast catabolism of OA that can be inferred from previous studies[25,38]. No detectable change in TA levels was observed (Supplementary Fig. 3b, c). *Tbh>Tbh* males did, however, show a small but significant increase in lifespan (Fig. 3c, Supplementary Fig. 3d), with no increase in median but a 5% increase in maximum lifespan (oldest 5%), indicating OA may be able to promote longevity in males. This effect was subsequently confirmed with OA feeding.

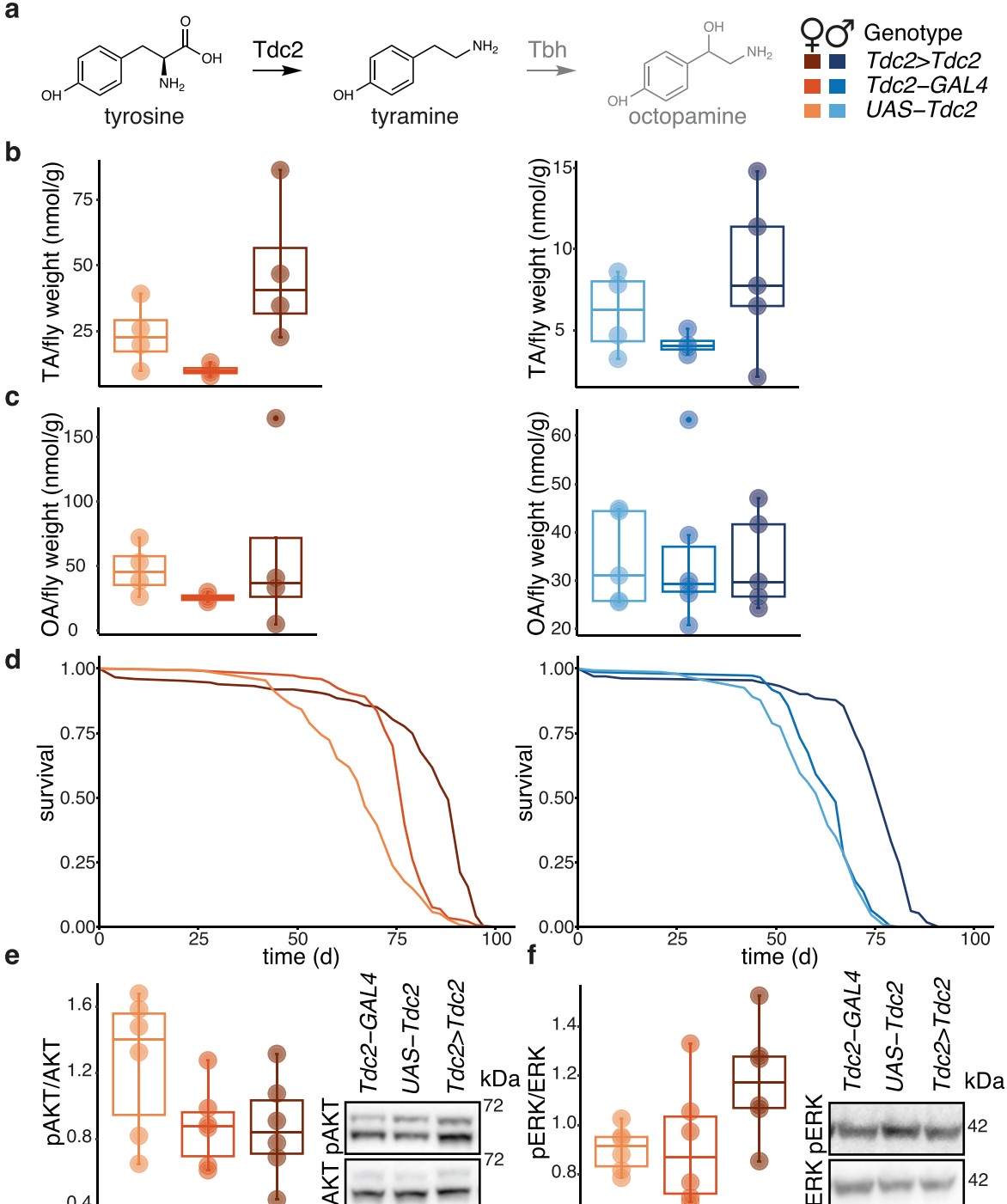

**Fig. 1 | Increased TA production in tyraminergic neurons promotes survival in old age. a** Biosynthesis of TA and OA with legend for subsequent panels: *Tdc2>Tdc2* refers to flies where *UAS-Tdc2* was induced in tyraminergic neurons with *Tdc2-GAL4*. **b**, **c** HPLC quantifications of TA and OA in *Tdc2>Tdc2* flies. TA: n = 4 biologically independent samples except for *Tdc2>Tdc2* males n = 5, OA: females n = 4, males n = 5 except for *Tdc2-GAL4* n = 6; TA: significant effect of genotype (driven versus controls p = 6.3 × 10⁻⁴), sex (p = 4.4 × 10⁻⁴) and genotype (driven versus controls)-by-sex interaction (p = 1.5 × 10⁻²), *Linear Model* (*LM*); OA: no significant effects, *LM*. **d** Lifespan of *Tdc2>Tdc2* flies. *Tdc2>Tdc2*: n = 144 dead/4 censored females and n = 122/10 males, *UAS-Tdc2*: n = 150/2 females, n = 135/12 males, *Tdc2-GAL4*: n = 146/1

females, n = 137/9 males; significant effect of genotype (driven versus controls p = 2 × 10⁻¹⁶) and sex (p = 2 × 10⁻¹⁶), *Cox Proportional Hazards* (*CPH*). **e** Levels of phosphorylated relative to total AKT (pAKT/AKT) and **f** pERK/ERK in *Tdc2>Tdc2* female flies. Quantification and representative blot images with an approximate position of a molecular weight marker are indicated. n = 6 biologically independent samples; pAKT/AKT: no significant effects, *LM*; pERK/ERK: significant effect of genotype (driven versus controls p = 0.02), *LM*. Detailed statistical analyses for (**b** to **f**) are shown in Supplementary Fig. 1a–c and 1k, respectively. **b**,**c**, **e**, and **f** - boxplots show quantiles with individual data points overlaid. Where relevant, statistical tests were two-sided; no multiple testing correction was applied.

## Orally administered TA extends lifespan and improves health in females

The smaller size of *Tdc2>Tdc2* flies indicated an impact on development from the constitutive, *Tdc2* induction in tyraminergic neurons. We wanted to examine the effects of increased TA and OA solely in adulthood. TA and OA administration in food is a commonly used experimental paradigm in *Drosophila*. TA and OA feeding can reverse several phenotypes associated with impaired ability to produce these bioamines, including behavioural and reproductive phenotypes[18,25,39,40], which indicates that TA and OA are taken up by the fly and can reach distal organs in an active form. We administered the two compounds orally from day two of adulthood, at a range of doses similar to those used previously[18,25]. Feeding 100 µg/ml TA in food resulted in increased levels of TA in flies of both sexes within 7 days (Fig. 4a, Supplementary Fig. 4a). A slight increase in OA was detected in males only (Fig. 4b, Supplementary Fig. 4b). TA feeding resulted in a significant, dose-dependent increase in female lifespan, with the longest extension observed on 100 µg/ml TA (Fig. 4c, Supplementary Fig. 4c, d). TA-fed females were not smaller, did not lay fewer eggs or feed less than the controls (Supplementary Fig. 4e–g), indicating the anti-ageing effects of TA can be separated from these phenotypes observed in *Tdc2>Tdc2* females, possibly by post-developmental or non-neuronal administration.

The extended lifespan in TA-fed females prompted us to also assess their health in older age. Feeding females with 100 µg/ml TA in food resulted in slightly reduced climbing performance in young flies and a delay in the age-related decline in climbing ability (Fig. 4d). The significant impact of TA on age-related decline was confirmed with a *LM* (significant age-by-TA feeding interaction, Supplementary Fig. 4h). Hence, the effects of TA feeding on climbing ability were broadly similar to those observed in *Tdc2>Tdc2* females but of reduced magnitude. Additionally, TA feeding resulted in fewer flies with a leaky gut across different ages (Fig. 4e, Supplementary Fig. 4i), indicating an improved gut barrier function. Hence, TA feeding from day two of adulthood was sufficient to improve both neuromuscular and digestive health in females.

Flies fed 25 µg/ml OA displayed an increase in the levels of OA in both sexes, albeit marginally so in males (Fig. 4b, Supplementary Fig. 4b), with no effect on TA levels (Fig. 4a, Supplementary Fig. 4a). A small but significant lifespan extension could be observed in males at this dose of OA (Fig. 4f, Supplementary Fig. 4j and k), similarly to what we observed for *Tbh>Tbh* males. Overall, adult-onset administration of TA/OA further corroborated their positive effects on longevity, additionally demonstrating their effectiveness after development. The two compounds showed distinct sexual dimorphism: TA robustly extended lifespan and improved health in old age in females, and OA had a marginally beneficial effect on male lifespan.

## Boosting intestinal β-adrenergic signalling promotes female longevity

To examine the signalling pathways that may act downstream of TA to promote female longevity, we focused on the gut due to its important role in ageing specifically in females[41], the sexually dimorphic longevity effects of TA feeding and the improvements in gut health observed in *Tdc2>Tdc2* and TA-fed flies. The fly genome encodes several receptors for TA and/or OA whose expression varies across tissues[42]. We focused on *Octß2R* and *TyrR*, as published single-cell expression data[43] indicated they are abundant in a number of cell types of the gut (Fig. 5a); their presence in the gut is corroborated by other techniques[42].

*Octß2R* encodes one of three fly receptors related to mammalian β-adrenergic receptors[42,44]. It is activated by both OA and TA, with a higher affinity for OA[44]. To manipulate the levels of *Octß2R*, we used the inducible GeneSwitch driver *TIGS*. This driver has been thoroughly characterised and it allows for expression restricted to the gut and induced by feeding of the RU486 inducer[45]. Within the gut, the

expression can be observed in both enterocytes and the ISCs[46,47]. In all experiments with inducible drivers, we started the induction from day two of adulthood so as to avoid any developmental effects of the manipulation. We found that adult-onset, gut-restricted induction of *Octß2R* was sufficient to extend lifespan, with a substantially stronger effect in females (Fig. 5b, Supplementary Fig. 5a), while the RU486 inducer had no significant effect on the survival of the driver-alone control (Supplementary Fig. 5b). The beneficial effects of *Octß2R* appeared to arise from the gut with some specificity, as increased expression of this receptor in the fat body (equivalent of adipose and liver) failed to substantially impact longevity (Supplementary Fig. 5c, using the *S106* driver which drives expression in the abdominal fat body with some expression detectible in the gut[45]). Additionally, gut induction of *Octß2R* had no effect on feeding, fecundity or weight (Supplementary Fig. 5d). On the other hand, gut-restricted induction of *TyrR*, a receptor principally responsive to TA and more distantly related to mammalian α- and β-adrenergic receptors[17,48], did not extend female lifespan (Supplementary Fig. 6a), and neither did the increased expression of its downstream effector, calcium/calmodulin-dependent protein kinase *CamKII*[49] (Supplementary Fig. 6b).

ß-adrenergic-like receptors, such as Octß2R, trigger an increase in the secondary messenger cAMP and the consequent activation of Protein Kinase A (PKA)[6,8,44]. Indeed, we found that increased expression of an activated form of a PKA catalytic subunit (*PKA\**, originally named mC\*, ref. 50) in the adult gut, using the *TIGS* driver combined with RU486 feeding from day two of adulthood, extended fly lifespan (Fig. 5c, Supplementary Fig. 6c, d). Stronger effects were again observed in females. Similarly to *Octß2R*, *PKA\** did not show beneficial effects when induced in other organs or cell types, namely the fat body or neurons (Supplementary Fig. 6e–g; using *S106* to drive expression in the fat and *elavGS* to drive expression in neurons[45,51]; neuronal induction of *PKA\** was detrimental). Lastly, to assess the cell types in which PKA activation was beneficial, we induced *PKA\** using the inducible *Mex1GS* or *GS5961* drivers. *Mex1GS* was generated using the *Mex1* promoter[52], which is predominantly expressed in enterocytes, while *GS5961* drives expression in the ISCs[34,53]. Only *Mex1GS > PKA\** females were long-lived upon RU486 feeding (Fig. 5d, e), indicating that activation within enterocytes is required for longevity. Lifespan extension was not observed in males and RU486 had no effect on the longevity of the driver-alone control (Supplementary Fig. 6h). Overall, our data are consistent with the activation of β-adrenergic-like signalling specifically in adulthood and in the female fly gut promoting longevity.

## *CrebB* links TA to longevity-promoting transcriptional programmes

The effects of TA on the female gut are likely to be paracrine or endocrine, as TA-producing neurons have not been noted to innervate the *Drosophila* gut[54,55]. Indeed, we observed the expression of membrane-bound GFP in the brain but not the gut of *Tdc2 > mCD8-GFP* females (Supplementary Fig. 7). Furthermore, local, non-neuronal Tdc activity is provided by the *Tdc2* paralogue, *Tdc1*, whose expression has been observed in gut muscle[19]; as confirmed in published single-cell expression data[43]. To provide further evidence of a potential endocrine role for TA, we queried its presence in haemolymph isolated from adult females using liquid chromatography coupled with tandem mass spectrometry (LC-MS/MS), based on a published protocol[56]. Ion trap MS/MS detected the accurate mass of [M + H]+ ion m/z for TA at 138.0913 and a m/z transition characteristic of TA (m/z 138 → 121) in haemolymph samples of untreated five day-old females (Fig. 6a). Using the same method, we observed an increase in haemolymph TA levels after two days of feeding with food containing 100 µg/ml TA (Fig. 6b).

Longevity achieved by alterations in endocrine signals has been noted to require transcriptional remodelling within the cells of the receiving tissue[57–60]. To understand the transcriptional changes triggered by TA and how they may be mediated, we profiled gene

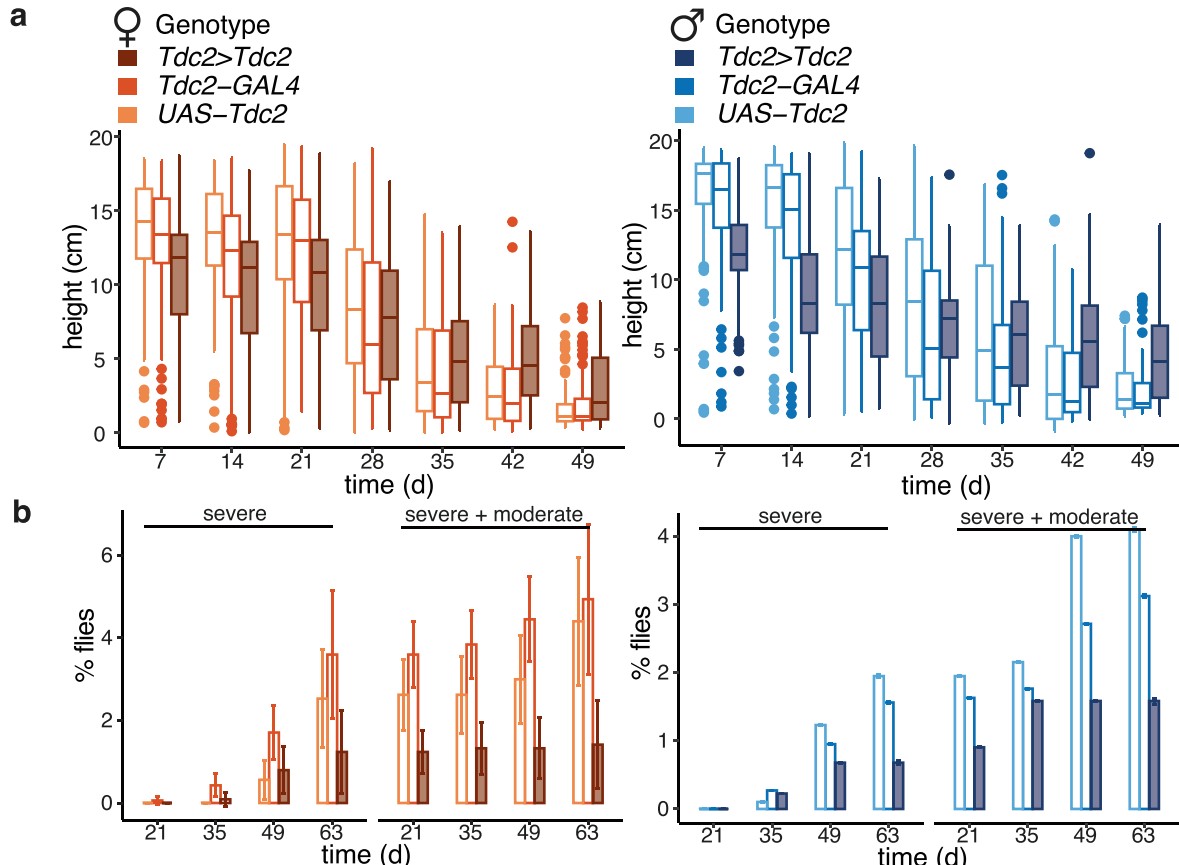

**Fig. 2 | Neuronal TA production promotes health in old age.** *UAS-Tdc2* was induced in tyraminergic neurons with *Tdc2-GAL4*. The legend for both panels is given in (**a**). **a** Height climbed by *Tdc2>Tdc2* flies. *Tdc2>Tdc2*: n/day=96/7, 85/14, 85/21, 80/28, 77/35, 86/42, 85/49 females, 80/7, 60/14, 64/21, 55/28, 54/35, 51/42, 54/49 males; *UAS-Tdc2*: n/day=112/7, 111/14, 108/21, 86/28, 98/35, 98/42, 94/49 females, 134/7,123/14,101/21,91/28, 67/35, 89/42, 77/49 males; *Tdc2-GAL4*: n/day=103/7, 99/14, 106/21, 95/28, 96/35, 95/42, 92/49 females, 119/7, 109/14, 88/21, 71/28, 87/35, 89/42, 77/49 males. Significant effect of genotype (driven versus controls, p = 2 x 10$^{-16}$), age (p = 2 x 10$^{-16}$), sex (p = 9.14 x 10$^{-6}$) and genotype (driven versus controls)-by-age interaction (p = 2 x 10$^{-16}$), *LM*. **b** Percentage of *Tdc2>Tdc2* flies with severe and severe plus moderate loss of gut barrier function (Smurf phenotype). *Tdc2>Tdc2*: n/day=435/21, 336/35, 240/49, 121/63 females, 190/21, 155/35, 89/49, 8/63 males; *UAS-Tdc2*: n/day=345/21, 294/35, 254/49, 175/63 females, 351/21, 334/35, 240/49, 50/63 males; *Tdc2-GAL4*: n/day=554/21, 542/35, 402/49, 143/63 females, 554/21, 538/35, 279/49, 92/63 males. Significant effect of age (p = 2 x 10$^{-16}$), sex (p = 0.03) and genotype (driven versus controls)-by-age interaction (p = 8 x 10$^{-3}$), *ordinal logistic regression*. Detailed statistical analyses for (**a**) and (**b**) are shown in Supplementary Fig. 2a, b, respectively. **a** - boxplot shows quantiles with individual data points overlayed; **b** - mean ± standard error of the mean (SEM). Where relevant, statistical tests were two-sided; no multiple testing correction was applied.

expression in the guts of flies fed TA using RNA-Seq (for full analysis results see Source Data). Consistent with its effect on lifespan, the transcriptional changes in response to TA were sexually dimorphic (Fig. 6c), with 128 genes differentially expressed in response to TA in females and 260 in males at 10% false discovery rate (FDR).

cAMP Responsive Element Binding (CREB) family of transcription factors regulate gene expression downstream of PKA and β-adrenergic signalling[61]. To query the involvement of the *Drosophila* CREB ortho-logue (CrebB) in the gut response to TA, we over-expressed *CrebB* in the adult gut and compared the sets of differentially expressed genes to those responsive to TA. In the case of females, we found a small but significant overlap (Fig. 6d) and the expression of transcripts within the overlap, such as the *dRNF34 (CG17019)* encoding an E3 ubiquitin ligase, was correlated between the two conditions (Fig. 6e). For males, the overlap was also small and significant but the expression of the genes within the overlap showed no significant correlation (Supplementary Fig. 8a). Consistent with this, re-analysing a published RNA-Seq dataset of gene-expression changes triggered in the gut by the induction of the CrebB partner CRTC[62], we additionally observed a significant overlap with gene expression changes triggered by TA in female guts (Supplementary Fig. 8b).

To directly test if CrebB transcriptional activity is stimulated by TA, we used flies carrying a cAMP Responsive Element (CRE) -

luciferase reporter, whose transcription is activated by CrebB, and fed them with TA for two weeks. Reporter expression was increased in both male and female flies after two days of TA feeding but after 15 days this activation was only sustained in females (Fig. 6f, Supplementary Fig. 8c), indicating that longer-term activation of CrebB only occurs in females. In females, we could also detect an increase in *CRE-luc* expression specifically in the gut after two days of TA feeding (Fig. 6g), indicating CrebB is activated by TA in this organ. Assessing the Gene Ontology (GO) categories that are enriched within the transcripts differentially regulated by *CrebB* in the female gut, we identified several processes whose activity in this organ was previously linked to longevity, such as proteolysis and cytoplasmic translation[47,63,64], as well as genes characterised as determining adult lifespan (Supplementary Fig. 8d). Overall, our observations were consistent with CrebB acting downstream of TA to promote transcription of known longevity-promoting genes and processes in the female gut.

### TA, intestinal *Octβ2R* and *CrebB* act within the same longevity pathway in females

Finally, we examined the interactions between longevity interventions involving TA, intestinal *Octβ2R*, and *CrebB* to ascertain whether they all function in the same longevity pathway in females. Firstly, we found that knocking down *CrebB* with RNAi specifically in the adult female

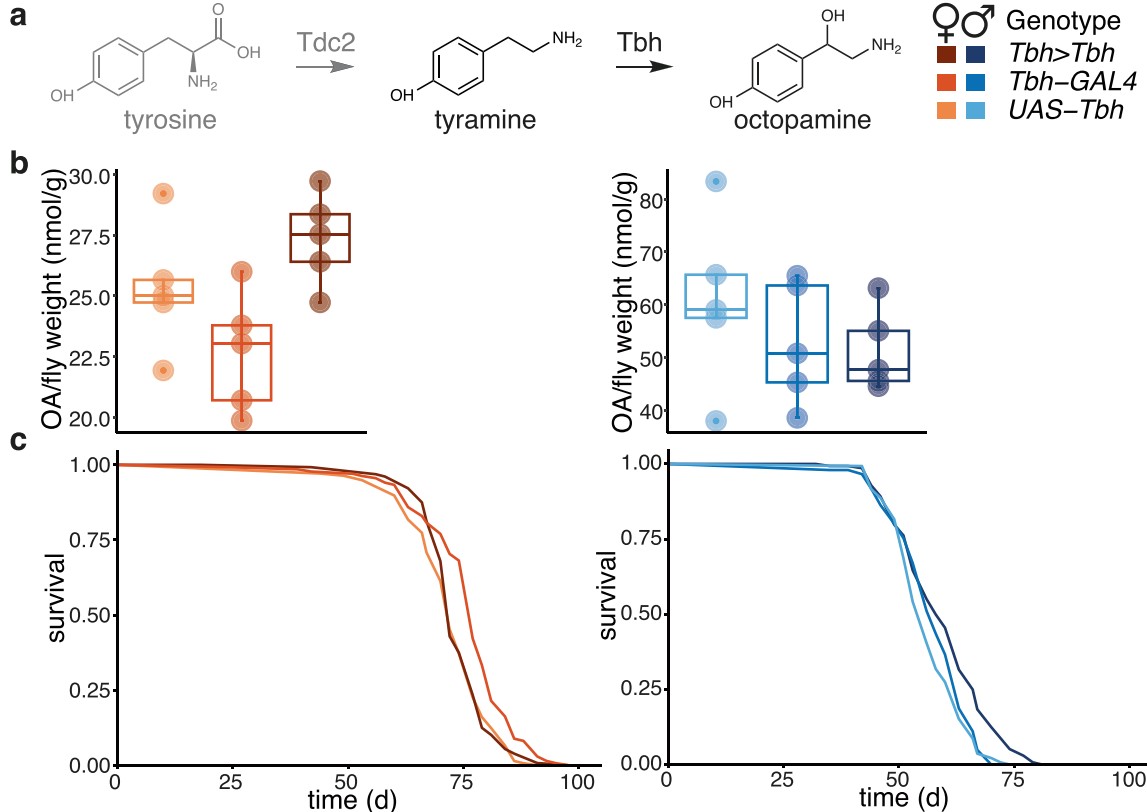

**Fig. 3 | Neuronal OA modestly increases male lifespan. a** Biosynthesis of TA and OA and the legend for subsequent panels. *Tbh>Tbh* refers to flies where *UAS-Tbh* was induced with *Tbh-GAL4*. **b** HPLC quantifications of OA in *Tbh>Tbh* flies. n = 5 biologically independent samples; significant effect of sex (p = 3.04 x 10⁻⁹), *LM*. Boxplots show quantiles with individual data points overlayed. **c** Lifespan of *Tbh>Tbh* flies. *Tbh>Tbh*: n = 128 dead/2 censored females and n = 137/7 males, *UAS-*

*Tbh*: n = 137/1 females, n = 139/4 males, *Tbh-GAL4*: n = 135/0 females, n = 145/3 males, significant effect of sex (p = 2 x 10⁻¹⁶), and genotype(driven versus controls)-by-sex interaction (p = 3.53 × 10⁻⁸), *CPH*. Detailed statistical analyses for (**b**) and (**c**) are shown in Supplementary Fig. 3a, d. Where relevant, statistical tests were two-sided; no multiple testing correction was applied.

gut was sufficient to abrogate the lifespan response to TA (Fig. 7a). To robustly test the statistical significance of this effect, we used *Cox Proportional Hazards* analysis (*CPH*), which confirmed that the lifespan response to TA was significantly altered by gut-restricted induction of RNAi against *CrebB* (significant RU486-by-TA feeding interaction, Supplementary Fig. 8e). Conversely, *CrebB* expression in the adult gut, using the *TIGS* driver, was sufficient to extend female lifespan (Fig. 7b); expression in other cell types, such as fat body cells or neurons (using *LSP2GS* to drive in the fat body body[65] and *elavGS* for neurons), or in males did not have such an effect (Supplementary Fig. 8f; neuronal expression was detrimental in males). *CrebB* induction was also sufficient to reduce age-related gut dysplasia (Supplementary Fig. 8g). Within the gut, expression in enterocytes, using an enterocyte-specific driver (*GS5966*; ref. 53), but not in ISC using *GS5961*, was sufficient for longevity in females (Fig. 7c, Supplementary Fig. 8h), recapitulating the cell-type specificity of longevity caused by PKA activation. There was no effect of RU486 on lifespan in males or driver-alone controls (Supplementary Fig. 8i-j). Note that the longevity effects of *Octß2R* and *PKA* across experimental repeats were on average larger than the effects of *CrebB* (Supplementary Fig. 9a), possibly indicating that the receptor and the kinase are able to engage additional pro-longevity factors.

Gut-specific *CrebB* knockdown resulted in substantially shortened female lifespan (Fig. 7a). On one hand, such a short lifespan indicated *CrebB* is required for normal gut function, but on the other, it raised the possibility that TA may simply not be able to remedy the detrimental effects of *CrebB* loss of function. To circumvent this issue, we

additionally tested whether manipulations that are individually beneficial show additive effects when combined; non-additive effects indicate the interventions act within the same longevity pathway[47,66–68]. We found that the induction of *Octß2R* and *CrebB* in adult fly gut did not have additive effects on lifespan (Fig. 7d). *CPH* analysis confirmed that the response to RU486 was different between the driver alone (no response) and the *TIGS>Octß2R*, *TIGS>CrebB* and *TIGS>Octß2R & CrebB* females, while there was no difference in the RU486 response between the *TIGS>Octß2R & CrebB* and *TIGS>Octß2R*, *TIGS>CrebB* (Supplementary Fig. 9b), indicating that *Octß2R* and *CrebB* act in the same longevity pathway in the gut. Similarly, induction of *Octß2R* in the adult female gut and TA feeding were not additive for lifespan (Fig. 7e), again confirmed by *CPH* analysis (significant RU486-by-TA feeding interaction, Supplementary Fig. 9c). Lastly, we tested whether knocking down *Octß2R* had an impact on the ability of TA to promote longevity (Fig. 7f). In *TIGS>Octß2 Rᴿᴺᴬⁱ* females, TA had a significant effect on lifespan in the absence of the RU486 inducer (p = 0.002, *log-rank test*) while its effect on survival was not significant in the presence of RU486 (p = 0.8, *log-rank test*). Indeed, *CPH* analysis confirmed that the response to TA was modulated by *Octß2R* knockdown (RU486-by-TA feeding interaction p = 0.03, Supplementary Fig. 9d). Note that the effect of *Octß2Rᴿᴺᴬⁱ* observed is likely limited by the incomplete knock down of *Octb2R* mRNA (Supplementary Fig. 9e, also reported in ref. 69).

In summary, we find that CrebB is activated by TA feeding, required for the longevity effect of TA feeding, and sufficient to extend lifespan; additionally, we find that intestinal *Octß2R* acts in the same

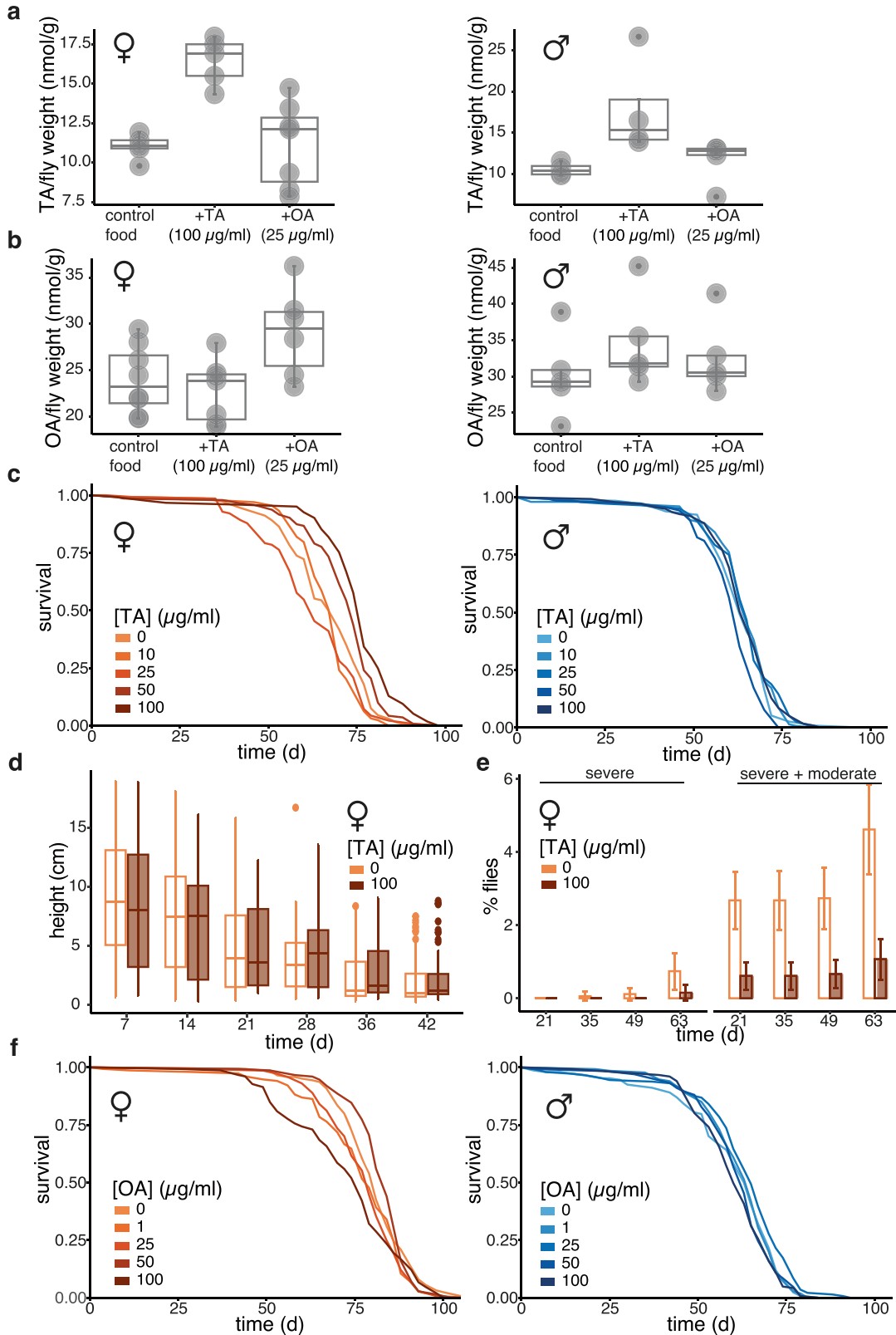

longevity pathway as orally administered TA and intestinal *CrebB*. Hence, our data are consistent with a longevity pathway in *Drosophila* females where TA can act on *ß*-adrenergic-like signalling in the gut to activate CrebB and thus extend lifespan. Additional pathways may exist in males responding to OA (and possibly TA) to promote longevity.

## Discussion

*ß*-adrenergic signalling has long been implicated in human age-related diseases, most notably cardiovascular disease. In heart failure, a dysregulation of *ß*-adrenergic signalling, including an early loss of *ß*-adrenergic responsiveness, is a key feature of disease pathophysiology[12]. Furthermore, a decrease in *ß*-adrenergic signalling

**Fig. 4 | TA feeding promotes female longevity.** TA and OA were fed to flies at indicated concentrations from day two of adulthood. **a** HPLC quantifications of TA in TA- and OA-fed flies. n = 5 biologically independent samples except for OA-fed females n = 7, control and TA-fed males n = 4; significant effect of TA feeding (p = 1.5 x 10⁻³) and TA feeding-by-sex interaction (p = 1.5 x 10⁻²), *LM*. **b** HPLC quantifications of OA in TA- or OA-fed flies. n = 5 biologically independent samples for males, females: control n = 8, OA-fed n = 6, TA-fed n = 7; significant effect of OA feeding (p = 1.8 x 10⁻²), sex (p = 1 x 10⁻⁴), TA feeding-by-sex interaction (p = 4.7 x 10⁻²), *LM*. Boxplots show quantiles with individual data points overlayed. **c** Lifespan of TA-fed flies. Control: n = 137 dead/10 censored females and n = 118/14 males; 10 μg/ml TA: n = 131/3 females and n = 145/8 males; 25 μg/ml TA: n = 139/2 females and n = 132/13 males; 50 μg/ml TA: n = 142/2 females and n = 120/21 males; 100 μg/ml TA: n = 119/3 females and n = 122/19 males; significant effect of TA feeding (p = 9.12 x 10⁻¹⁶), sex (p = 3.92 x 10⁻⁵), and TA feeding-by-sex interaction (p = 3.10 x 10⁻¹⁰), *CPH*. **d** Height climbed by TA-fed female flies. Control: n/day=74/7, 68/14, 74/21, 80/28, 76/35, 76/

42; TA-fed: n/day=75/7, 70/14, 67/21, 83/28, 82/35, 73/42. Significant effect of age (p = 2 x 10⁻¹⁶) and TA-by-age interaction (*p = 0.036*), *LM*. **e** Percentage of TA-fed female flies with severe and severe plus moderate loss of gut barrier function (Smurf phenotype) presented as mean ± SEM. Control: n/day=427/21, 395/35, 381/49, 295/63; TA-fed: n/day=443/21, 436/35, 423/49, 340/63. Significant effect of food (*p = 1.19x10⁻⁶*) and age (*p = 8.22x10⁻¹¹*), *ordinal logistic regression*. **f** Lifespan of OA-fed flies. Control: n = 138/9 females and n = 136/13 males; 1 μg/ml OA: n = 153/0 females and n = 116/18 males; 25 μg/ml OA: n = 142/3 females and n = 144/3 males; 50 μg/ml OA: n = 149/1 females and n = 131/4 males; 100 μg/ml OA: n = 151/1 females and n = 147/4 males, significant effect of OA feeding (p = 4 × 10⁻²), sex (p = 2 x 10⁻¹⁶), and OA feeding-by-sex interaction (*p = 1x10⁻⁴*); *CPH* using data for concentrations up to and including 25 μg/ml. Detailed statistical analyses for (**a** to **f**) are shown in Supplementary Fig. 4a–c, 4h–j, respectively. Where relevant, statistical tests were two-sided; no multiple testing correction was applied.

has been implicated in obesity[8]. For example, macrophages able to degrade noradrenalin produced by the sympathetic nervous system within the adipose tissue drive an age-related decrease in β-adrenergic signalling and lipolysis[70,71]. Indeed, promoting β-adrenergic signalling can yield beneficial metabolic effects in older mice[72]. Our study indicates that tissue-specific activation of the β-adrenergic-like pathway in adult fruit flies can promote whole organism longevity. Interestingly, the beneficial effects arise from the gut; adrenergic neurons that innervate the mammalian gut regulate the regeneration of the gut epithelium through β-adrenergic signalling[9], making it tempting to speculate that boosting the β-adrenergic pathway in this organ may also promote mammalian longevity.

Our findings are consistent with the ligand, TA, engaging the β-adrenergic pathway in the gut. It is interesting that the receptor implicated, *Octβ2R*, is responsive to both TA and OA[44], and yet OA does not have the same effect longevity-promoting effect in females as TA. This may not be due to the receptor specificity but rather to the in vivo, local availability of the two compounds and the way they can sustain the activation of the downstream pathway. Indeed, TA is produced in the gut muscle[19] and likely to be an important regulator of gut physiology; this role for TA is as yet to be fully characterised.

We detect TA in heamolymph, indicating that it may act in an endocrine fashion, a mode of action that has received very little attention in the fruit fly. An additional source of TA appears to be the microbiota, and in the context of high-fat feeding, microbiota-derived TA can protect against diet-induced insulin resistance acting though TyrR, intracellular Ca²⁺ and CrebB[73]. The source of TA can make a difference to the physiological outcome: we find that over-expression of *Tdc2* in tyraminergic neurons can extend male lifespan, while TA feeding cannot. This may be due to specific effect of TA directly at sites innervated by tyraminergic neurons; similarly, some deficits in neuronal OA production cannot be rescued by OA feeding[37]. Neuronally produced TA may also have beneficial effects beyond those achievable by TA feeding even in females, as both health and lifespan appear more strongly altered in *Tdc2>Tdc2* females than in females fed TA. Some of these effects may be indirect e.g. by alterations in activity or other behaviours, or caused by increased TA during development; additionally it is conceivable that some may be due to functions of tyraminergic neurons that are independent of TA. Still, regardless of the source of TA, our data show that engaging the β-adrenergic-like signalling pathway in the adult *Drosophila* gut is sufficient to promote female longevity. Importantly, our findings describe a *potential* for the pathway to improve health by increasing its activity beyond that of the otherwise healthy, wild-type fly; this is consistent with the evolutionary understanding of ageing as arising beyond the reach of natural selection that shapes wild-type physiology of a species[74]. The longevity pathway we describe culminates in the activation of the transcription factor CrebB that drives an increase in expression of genes counteracting ageing, including those within the proteostasis network, a

cellular network implicated in longevity[75] and suspected to be regulated by CrebB[62]. Hence, CrebB emerges as a link between the β-adrenergic-like endocrine signals and a longevity-promoting transcriptional programme.

The beneficial effects we observe are sexually dimorphic. This aligns well with other detailed investigations of TA and OA biology, which are increasingly appreciating the sexual dimorphisms within this neuroendocrine axes: from the existence of sexually dimorphic neuroanatomy underlying behaviours[76], to the mechanistic underpinnings of sexually dimorphic responses to exercise[25]. The beneficial effects of TA observed in females are consistent with TA acting on the fly gut, as the physiology of this organ is strongly shaped by the sex of the animal so that ageing has a more profound, detrimental effect on its function in females[41,77,78], explaining the sexual dimorphism we observe. Such sexual dimorphism is frequent in longevity, but it cannot be mapped easily between species. For example, inhibition of the TORC1 or MEK kinases promotes female longevity in flies[66,79] while it provides benefits in both sexes in mice[80,81]. Hence, it is possible for localised β-adrenergic stimulation to boost longevity in both sexes in mammals. The mechanism behind the minor, beneficial effects of OA on ageing observed in males remains to be studied but the sexual dimorphism in this case is reminiscent of the sex-specific impact of exercise, which is mediated by OA[25].

Physical exercise is one of the most widely promoted, and possibly most effective, simple measures that can delay multiple facets of human ageing[82]. Yet, the biological mechanisms mediating this anti-ageing effect remain unclear. Adrenergic signalling both facilitates exercise and is shaped by it[83]. Indeed, recent efforts at investigating the mechanism behind the health-promoting effects of exercise in rodents have revealed that exercise profoundly shapes gene expression in the adrenal gland[84], and that these changes can be genetically linked to human complex traits[85]. Even in flies, OA acting though adrenergic-like receptors drives exercise adaptations specifically in males[25,69]. Interestingly, our work shows that this is the sex to which OA confers longevity. Our study suggests that remodelling of adrenergic signalling may underlie aspects of the anti-ageing effects of physical exercise and further investigation may reveal tissues and molecular mechanisms through which this neuroendocrine pathways links exercise and longevity.

## Methods

### Fly stocks and husbandry
The outbred wild-type stock was obtained in present-day Benin in 1970s and has been maintained in population cages to preserve fecundity and lifespan at levels of freshly caught flies. The *w¹¹¹⁸* or *v¹* mutation mutation was incorporated into this background by backcrossing to allow tracking of genetic constructs. Outbred *w¹¹¹⁸* population was cleared of *Wolbachia* by tetracycline treatment several years ago; all experimental flies were negative for *Wolbachia*. Transgenes:

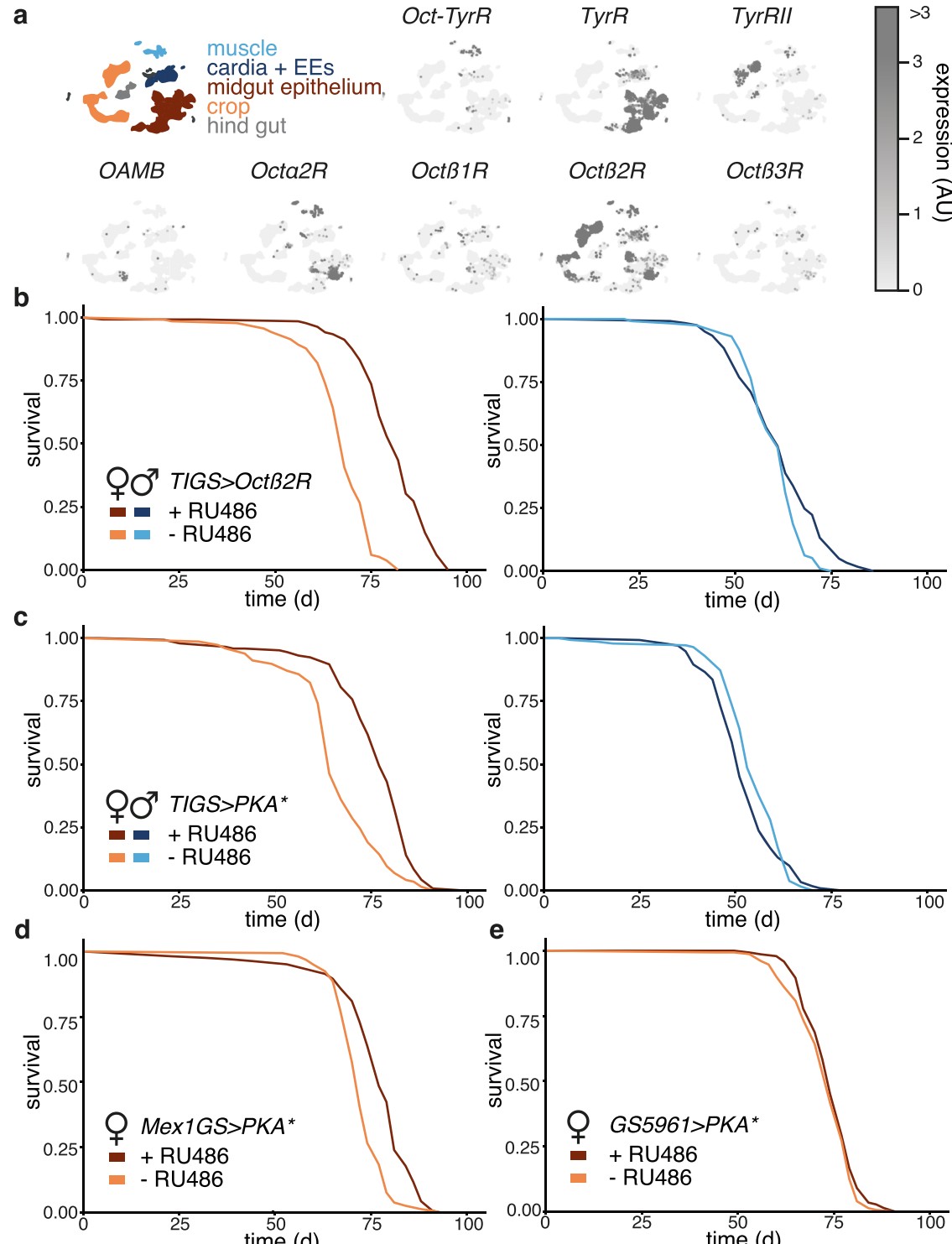

**Fig. 5 | ß-adrenergic-like signalling in the female gut promotes longevity.**
**a** Single cell expression of different TA and OA receptors in the fruit fly gut (data replotted from Fly Single Cell Atlas). Simplified annotation is given on the left. **b** Lifespan of *TIGS>Octβ2R* flies. *Octβ2R* was induced in the gut upon RU486 feeding. Induced (plus RU486): n = 136 dead/5 censored females and 121/0 males; uninduced (minus RU486): n = 136/4 females and 112/7 males; significant effect of RU486 (p = 2 x 10⁻¹⁶), sex (p = 3.65 × 10⁻¹³), and RU486-by-sex interaction (p = 9.84x10⁻⁸), *CPH*. **c** Lifespan of *TIGS > PKA\** flies. *PKA\** was induced in the gut upon RU486 feeding. Induced (plus RU486): n = 144/3 females and 131/5 males; uninduced (minus RU486): n = 147/0 females and 137/7 males; significant

effect of RU486 (p = 1.7 x 10⁻⁴), sex (p = 2 × 10⁻¹⁶), and RU486-by-sex interaction (p = 2.4 x 10⁻³), *CPH*. **d** Lifespan of *Mex1GS > PKA\** female flies. *PKA\** was induced in enterocytes upon RU486 feeding. Induced (plus RU486): n = 163/0; uninduced (minus RU486): n = 158/0; effect of RU486 (p = 2.3x10⁻¹²), *log-rank test*. **e** Lifespan of *GS5961 > PKA\** female flies. *PKA\** was induced in ISCs upon feeding RU486. Induced (plus RU486): n = 144/1 females; uninduced (minus RU486): n = 151/1 females; effect of RU486 (p = 0.15); *log- rank test*. In (**b** to **e**), RU486 feeding started on day two of adulthood. Detailed statistical analyses for (**b**) and (**c**) are shown in Supplementary Figs. 5a, 6c, respectively. Where relevant, statistical tests were two-sided; no multiple testing correction was applied.

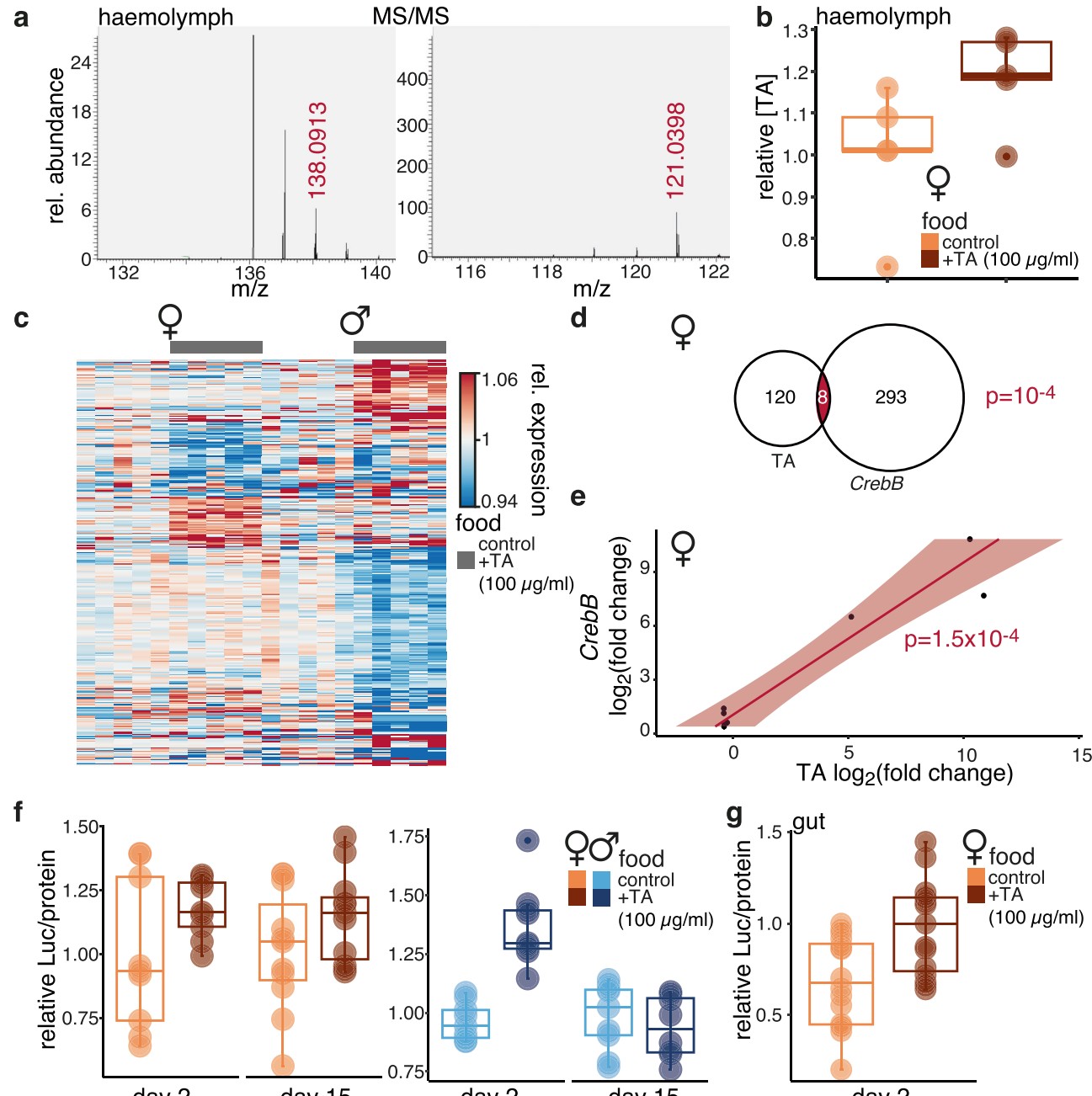

**Fig. 6 | TA activates *CrebB* in the female gut. a** LC-MS/MS (ion trap) showing mass of [M + H]+ ion m/z for TA at 138.0913 and a m/z transition characteristic of TA (m/z 138 → 121) in haemolymph samples of untreated young females. **b** LC-MS/MS quantifications of haemolymph TA in TA-fed and control female flies, relative to control. n = 5 biologically independent samples; effect of food (*p = 0.039, one-sided t-test*). **c** Heatmap of transcripts differentially expressed upon TA feeding in males or females. Expression data were transformed by variance stabilising transformation and normalised to control average. **d** Overlap of transcripts differentially expressed in the gut upon *CrebB* induction (in *TIGS>CrebB*) and TA feeding in females. *P* value from a *one-sided hypergeometric test*. **e** Correlation in expression upon TA feeding or *CrebB* induction for the genes within the overlap in **d**, showing the line of best fit and its 95% confidence interval (shaded). **f** Relative luciferase activity in *CRE-luciferase* flies after TA feeding for the indicated time (days on TA). n = 9 biologically independent samples except for females day 15 n = 11, TA-fed males day 15 n = 8; significant effect of TA feeding (*p = 0.037*), TA feeding-by-sex-by-time interaction (*p = 0.023*), *LM*. **g** Relative luciferase activity in the gut of *CRE-luciferase* flies after TA feeding for the indicated time. n = 15 TA-fed and 16 control biologically independent samples; significant effect of food (*p = 0.003, t-test*). In (**b** to **g**), TA feeding started at day two of adulthood. Detailed statistical analyses for (**f**) are shown in Supplementary Fig. 8c. **b**, **c**, and **g** - boxplots show quantiles with individual data points overlayed. Where relevant, statistical tests were two-sided unless otherwise noted; multiple testing correction was applied only in transcriptomic analyses and as indicated in the text.

---

*UAS-Tdc2* (P{UAS-Tdc2.C}, ref. 19, RRID:BDSC_9316), *UAS-Tbh* (ref. 35), *UAS-Octβ2R* (P{UAS-Octβ2R.S}, deposited in Bloomington Drosophila Stock Centre (BDSC) by Schwarzel, M., RRID:BDSC_78806), UAS-PKA* (*UAS-mc**, ref. 50)*, *UAS-TyrR* (P{UAS-TyrR.H}, ref. 86, RRID:BDSC_67128), *UAS-CaMKII* (P{UAS-CaMKII.R3}, ref. 87,

RRID:BDSC_29662), *UAS-CrebB* (P{UAS-CrebB-17A-a.cor}, ref. 88, RRID:BDSC_9233), 5xCRE-Luc (P{5xCRE-LUC}, deposited in BDSC by Fayyazuddin, A., used in ref. 62, RRID:BDSC_79016), *UAS-CrebB*^RNAi (P{TRiP.HMJ30249}, ref. 89, RRID:BDSC_63681), UAS-Octβ2R^RNAi (P{GD2954} from Vienna Drosophila Resource Center (VDRC), used in

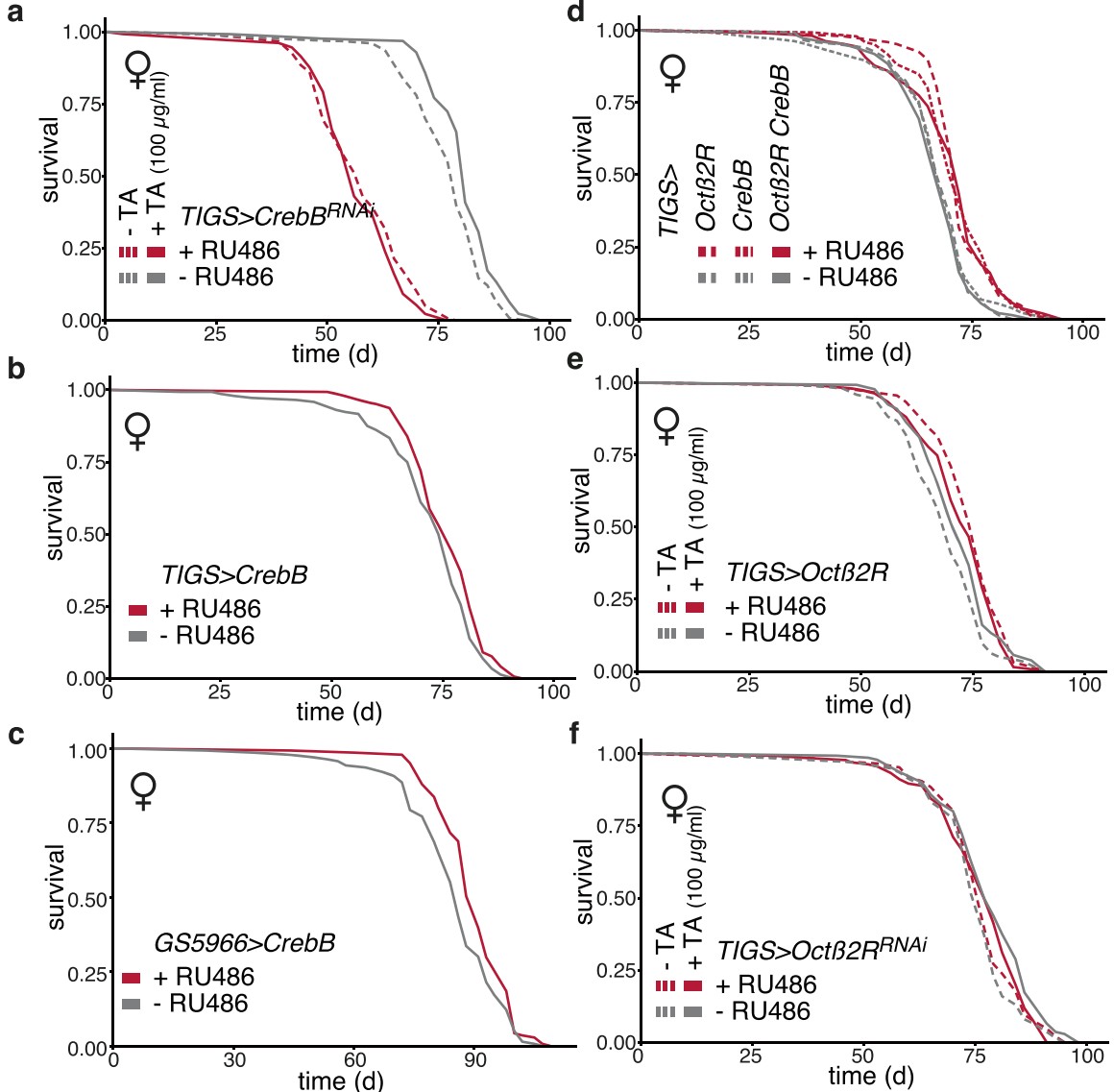

**Fig. 7 | Intestinal *CrebB* activity delays ageing. a** Lifespan of *TIGS>CrebB^{RNAi}* females. *CrebB^{RNAi}* was induced in the gut by RU486 and combined with TA feeding. Induced (plus RU486): n = 119 dead/3 censored; induced and TA-fed: n = 130/0; uninduced (minus RU486): n = 123/3, uninduced and TA-fed: n = 127/0; significant effect of RU486 ($p = 2 \times 10^{-16}$), TA feeding ($p = 1.5 \times 10^{-3}$), RU486-by-TA feeding interaction ($p = 1.2 \times 10^{-3}$), *CPH*. **b** Lifespan of *TIGS>CrebB* females. *CrebB* was induced in the gut. Induced (plus RU486): n = 143/0; uninduced (minus RU486): n = 144/1. Effect of RU486 ($p = 2.25 \times 10^{-2}$), *log-rank test*. **c** Lifespan of *GS5966>CrebB* female flies. *CrebB* was induced in enterocytes. Induced (plus RU486): n = 141/0; uninduced (minus RU486): n = 140/0; significant effect of RU486 ($p = 2.6 \times 10^{-3}$); *log-rank test*. **d** Lifespan of females with combined induction of *CrebB* and *Octβ2R* in the gut. *TIGS*-alone data are shown in the Supplementary Fig. 9b. *TIGS>CrebB & Octβ2R* induced (plus RU486): n = 106/1, uninduced (minus RU486): n = 147/0; *TIGS>CrebB* induced: n = 145/0, uninduced: n = 133/1; *TIGS>Octβ2R* induced: n = 148/0, uninduced: n = 149/1; significant effect of RU486 ($p = 3.6 \times 10^{-11}$) and RU486-by-genotype (for contrast of *TIGS* versus *TIGS>Octβ2R* and *TIGS>CrebB* and *TIGS>Octβ2R & CrebB*, $p = 7.9 \times 10^{-5}$), *CPH*. **e** Lifespan of *TIGS>Octβ2R* females. *Octβ2R* was induced in the gut and combined with TA feeding. Induced (plus RU486): n = 133/2; induced and TA-fed: n = 135/0; uninduced (minus RU486): n = 144/0, and uninduced and TA-fed: n = 127/6 censored; significant effect of RU486 ($p = 1.3 \times 10^{-5}$), TA feeding ($p = 2.4 \times 10^{-2}$), RU486-by-TA feeding interaction ($p = 1.5 \times 10^{-2}$), *CPH*. **f** Lifespan of *TiGS>Octβ2R^{RNAi}* flies upon induction with RU486 combined with TA feeding. Induced (plus RU486): n = 147/0; induced and TA-fed: n = 135/0; uninduced (minus RU486): n = 138/1, and uninduced and TA-fed: n = 139/3 censored; significant effect of TA feeding ($p = 0.001$), RU486-by-TA feeding interaction ($p = 0.03$), *CPH*. In all panels, TA and/or RU486 feeding started from day two of adulthood. Detailed statistical analyses for (**a**, **d**, **e** and **f**) are shown in Supplementary Figs. 8e, 9c–d respectively. Where relevant, statistical tests were two-sided; no multiple testing correction was applied.

ref. 69), *UAS-mCD8-GFP* and drivers: *Tdc2-GAL4* (P{Tdc2-GAL4.C}, ref. 19, RRID:BDSC_9313), *Tbh-GAL4* (P{GMR76H05-GAL4}, ref. 36, RRID:BDSC_45904), *TIGS* and *S₁106* (ref. 45), *GS5961* and *GS5966* (ref. 53), *Mex1GS* (ref. 52) *LSP2GS* (ref. 65) and *elavGS* (ref. 51) were backcrossed at least six times into the outbred background. The transgenes were kept in large populations and frequently outcrossed back into the outbred population to avoid bottlenecks and genetic drift. *UAS-CrebB* was tracked by PCR using the following primers:

forward, CGAGCGGAGACTCTAGCGAGC and reverse, GCAGTTGATT-TACTTGGTTGCTGG. Expression of *Tdc2-GAL4* in female brain and gut was assessed in *Tdc2 > mCD8-GFP* young adults after fixation in 4% paraformaldehyde in PBS for 15 minutes, washing three times, mounting in Vectashield and visualisation with a Zeiss Airyscan 880. For brains, single confocal planes were taken every 5 μm on z-stack and planes were stacked together. For guts, the images were taken using tile-scan. All settings were kept the same between the two images.

Stocks were maintained and experiment conducted at 25 °C, 60% humidity, and 12h:12h light/dark cycle on a yeast/sugar/agar (SYA) food medium[90]. All the crosses were performed in cages containing grape juice, agar, and live yeast. When using GeneSwitch drivers, RU486 (Sigma, dissolved in ethanol) was added to a final concentration of 200 μM to SYA food. TA and OA (Sigma-Aldrich, St. Louis, MO) were dissolved in $H_2O$ and added to SYA food at the noted concentrations. As control, the vehicle alone equivalent to the highest concentrations of TA/OA/RU486 was added. For Smurf and food consumption assays, blue dye (FD & C blue dye no. 1, Fastcolors) was added to SYA food to 2.5% (w/v).

### Generation of experimental flies
Eggs were collected from suitable crosses over <22 h, washed in 1% PBS, and seeded into bottles at 20 μl egg sediment per bottle to achieve standardised larval density. Adult flies emerged after 10 days, were transferred to new food, allowed 48h to mate, and then sorted into experimental vials by light anesthetization with CO2 at a density of 15 flies per vial in single-sex groups (~5 flies per vial for fecundity, proboscis extension, and weight assays). Thus, treatment with RU486, TA, or OA started on day two of adulthood. Experimental vials were kept in Drosoflippers (https://www.drosoflipper.com/).

### Lifespan, fecundity and body-weight assays
For lifespan assay, flies were transferred to new food and their survival scored three times a week until all flies died. For the fecundity assay, eggs laid over ~24 h at the indicated times were counted. To measure body weight, flies were snap-frozen in liquid nitrogen at indicated times and individual adult flies were weighed on a Mettler Toledo AT201 precision balance.

### Gut barrier function (Smurf) assay and pH3 staining
Smurf assays were performed to measure gut barrier function as follows: flies were transferred to Smurf food for 48 h before being scored as not Smurf if the blue dye had not leaked out of the gut, as partial Smurf if the blue dye had leaked out of the gut but had not reached the head, and as full Smurfs if completely blue. Immunostaining was performed essentially as described[64] with rabbit anti-phospho-H3 (Cell Signalling #9701). Stained guts were mounted in mounting medium in DAPI (Vectastain), and the number of pH3 positive cells in each midgut was counted using a fluorescent microscope.

### Climbing (negative geotaxis) assays
Climbing assay was performed as before[64]. Briefly, flies were transferred to empty vials at indicated times in Drosoflippers in such a way that they could climb a two-vial height, allowed to acclimatise for 30 min, then gently tipped to the bottom of the vial, and climbing was recorded immediately for 20 s. Video stills from the same time point (15 s; the time point when maximum height was reached by young wild-type flies) were analysed using ImageJ (https://imagej.nih.gov/ij/). Flies whose height could not be determined from the still were not included in the analysis.

### Feeding (proboscis extension and food consumption) assays
Fly feeding frequency was measured using proboscis extension assays as described previously[91]. In brief, the assays were performed two-hours after lights on, after the flies' overnight acclimatisation to the room and 30 min acclimatisation to the observers. Feeding events (proboscis extended to food) were scored for 3 seconds per vial consecutively every 10 min over 100 minutes in total. The feeding frequency was calculated as the sum of observed feeding events over the total number of feeding opportunities (number of flies in a vial x number of vials per condition x number of observations). The observers were blinded to the experimental conditions.

Food consumption was quantified using food containing a blue dye, with minor modifications from the previously described method[91]. Briefly, 15 flies were were allowed to feed on SYA food containing 2.5% (w/v) blue dye for 30 min, snap frozen in liquid nitrogen, homogenised in 500 μl of distilled water for 30 s at 6500 Hz in a ribolyser, extract cleared by cetrifugation and the absorbance of 200 μl aliquots was measured at 630 nm using a microplate reader. The absorbance measurements falling within the linear range was confirmed by serial dilution of blue dye in water.

### TA/OA HPLC quantification and LC-MS/MS
TA and OA were quantified with High Performance Liquid Chromatography (HPLC) on Nexera X2 HPLC (Shimadzu, Japan) equipped with a C18 column (150x4.6 mm, 3um particle size, Sigma Aldrich, St Louis, USA) based on a published method[24] with minor modifications. 20 adult flies were weighed immediately after flash-freezing in liquid nitrogen and homogenised in 200 μl extraction buffer (1:1 acetonitrile: methanol) by shaking twice with glass beads (Sigma G8772) for 30 s at 6500 Hz in a ribolyser, followed by centrifugation at $20000 \times g$ for 12 min. TA (Cat. No.#422380050, ThermoScientific) and OA (Cat. No.# 63681, Sigma Aldrich) standards (10μM, 1μM, and 100nM) were prepared in the same buffer. Samples were filtered and 20 μl were injected into the column followed by isocratic elution with HPLC-grade methanol:acetonitrile:10 mM sodium pentane pH3 (7.5:7.5:85,v/v/v) at 0.4 ml/min for 35 min. Fluorescence (280 nm excitation, 320 nm emission) was detected with RF-20Axs fluorescence detector. TA and OA peaks were confirmed by spike-in to select unknown samples and levels quantified relative to standards by measuring peak areas. The values were normalised with fly weight. Flies that were fed TA/OA were placed on food not containing TA/OA for 30 min before harvesting to clear their gut contents, as described[92].

1 μl of haemolymph was collected from 15 females by decapitation and mild centrifugation as described[93], after at least 30 min of allowing them to clear their gut content on food not containing TA, and diluted into 40 μl of the HPLC extraction buffer. TA was detected and quantified with LC-MS/MS system (Vanquish LC, Thermo Fisher Scientific, UK) coupled to a heated electrospray (HESI) probe and Q Exactive mass spectrometer (Thermo Fisher Scientific, UK). A 20 μL sample was injected onto a Hypersil GOLD reverse-phase column (100 mm x 2.1 mm, particle size 1.9 μm) at 300 μL/min flow rate. Mobile phase A was 0.1 % formic acid in water, and B was 0.1 % formic acid in acetonitrile. An eight-min gradient was: 1% B for 0.5 min, raised to 50% B over 5 min, held for 1 min, and then back to 1% B in 6 s, and held for 1.9 min to equilibrate the column. The eluent was directed to the HESI source of the Q Exactive mass spectrometer. The HESI probe was operated with a sheath gas flow at 25 arb, an auxiliary gas flow at 10 arb, a spray voltage at 3.5 kV, a capillary temperature at 320 °C, and an S-lens RF level at 55. The auxiliary gas heater was set to 50 °C. The Q Exactive mass spectrometer was operated in positive-ion mode with a collision energy of 35%.

### Luciferase assays
Luciferase activity was measured with the Firefly luciferase assay kit (Promega Cat# E152A). 4 adult flies were flash-frozen in liquid nitrogen, homogenised in 80μl cell lysis buffer by shaking with glass beads (as above), and lysates cleared by centrifugation at $18000 \times g$, 4 °C for 15minutes. For measurements of gut luciferase activity, the extract was obtained from 10 dissected guts using the same method. 10 μl of sample were assayed as per manufacturer's instructions on microplate reader (Infinite® M Plex, Tecan) and signal normalised to total protein concentrations determined using bicinchoninic acid (BCA) protein determination kit (Thermo Scientific, Cat#23225) with a BSA standard as per manufacturer's instructions. To protect against outliers, the two highest and two lowest measurements from each condition/time point were excluded from the analysis of whole-fly data.

## Western blotting

Total protein was extracted with trichloroacetic acid from five female flies and separated on gradient gels (Invitrogen, NP0322-BOX) in MES buffer. Proteins were transferred to nitrocellulose membranes (Amersham) for 30 minutes using a semi-dry transfer system (Trans-Blot Turbo, BioRad), blocked in 5% skimmed milk for 1 h at room temperature, and incubated overnight with one of the following primary antibodies (Cell Signalling Technology): anti-total-ERK (4695), anti-phospho-ERK (4370), anti-total-AKT (9272), or anti-phospho-AKT (9271) at 1:1000 in 5% BSA. Primary antibody binding was detected with a goat anti-rabbit-HRP secondary antibody (1:10,000, Abcam ab6721); the blots were developed with Immobilon Crescendo Western HRP substrate (Merck, WBLUR0500) and imaged using an Amersham ImageQuant800 imager. Chemiluminescence images (.tiff) were quantified with ImageJ (v1.54 d). A total of six replicates were analysed in two independent blots (uncropped images of the blots are given in Supplementary Fig. 10).

## RNA extraction, RNA sequencing and analysis

For RNA extraction, eight mid-guts from each condition or genotype were dissected at the indicated times in ice-cold PBS and placed into ice-cold TRIzol (Thermo Fisher Scientific, 15596026). These were 7-day-old, wild-type flies that were either fed control or TA-containing food, or 7-day-old *TIGS>CrebB* or *TIGS* alone flies that were fed control or RU486-containing food. Five experimental replicates were performed for each condition and RNA samples were preserved at −80 °C prior to extraction by TRIzol/choloroform method as per manufacturer's instructions and quantified by NanoDrop 2000c spectrophotometer. For RNA sequencing, library preparation (Illumina, strand-specific RNA-seq with PolyA selection) and sequencing (Illumina NovaSeq 2 x 150 bp sequencing) were performed by Azenta Life Sciences (GENEWIZ UK Ltd., Essex, UK). Read quality was assessed with FastQC[94] and transcript abundance was estimated with Salmon[95] against the fasta file containing all *Drosophila* transcripts (Flybase, r6.55). Data was imported into R[96] with RStudio, using *tximport* package[97] and differential expression assessed with *DESeq2*[98] and *ihw*[99] packages. Separate DESeq2 models were fitted for TA versus control food in males or females; RU486 versus control food in males or females, separately for *TIGS>CrebB* or *TIGS* alone flies. Any transcripts whose levels were significantly altered by RU486 in the driver alone control and in *TIGS>CrebB* flies in the same direction were removed from the results and subsequent analysis (13 for females and 4 for males). GO term enrichment was analysed using *TopGO*[100] and the heatmap was made from VST-transformed data that were normalised for the average of the control using *pheatmap* package in R. RNA-Seq data examining the effect of CTRC in fly gut[62] were analysed in the same way. Raw data have been deposited in Gene Expression Omnibus (GEO), accession numbers GSE276897 and GSE276880, and analysis results are given in Source Data.

## qPCR

RNA was extracted from female guts and converted to cDNA using SuperScript II Reverse Transcriptase (Thermo Fisher Scientific, 18064014) and Oligo-dT primers. qPCR was performed on an Applied Biosystems QuantStudio 6 Flex real-time PCR instrument with Fast SYBR Green PCR Master Mix (Applied Biosciences, 4385612) as described before[101]. Expression data were normalised to Actin and scaled for batch. Statistical analysis was performed using *t-test*. Actin primers were described[101], *Octβ2R* primers were:

 OctB2R-qRT_Forward: CCATCTTCATCGGCTGGTACA
 OctB2R -qRT_Reverse: CGGCGTAGTACTTGTTCACCA

## Statistical analyses

Statistical analyses were performed in R using the following functions or packages: *lm*, *glm*, *survival*, and *ordinal*. All regression analyses had a full-factorial design except for ordinal logistic regression, where a model with a full-factorial design was initially fitted and subsequently reduced to remove non-significant interaction terms. Tests were two-sided unless otherwise noted. Details of tests and results are reported in figure captions (significant effects only) and Supplementary Figs. (full results).

## Reporting summary

Further information on research design is available in the Nature Portfolio Reporting Summary linked to this article.

## Data availability

Raw RNA-Seq data have been submitted to GEO, accession numbers: GSE276897 and GSE276880. Data previously deposited in GEO was also used, accession number: GSE185159. Uncropped images of the western blots are available in Supplementary Fig. 10. Source data are provided with this paper.

## Code availability

No custom code was used in this study.

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

## Acknowledgements

We thank M. Monastirioti, J.-R. Martin, B. Ohlstein, S. Pletcher, and L. Partridge for fly lines; E. Mvuama Bunaisa, A. Blake, N. Markevych, and A. Nicheperovich for technical assistance; Dr. Kersti Karu at UCL Chemistry Mass Spectrometry Facility for her help with the operation of the LC-MS/MS instrument; T. Niccoli for comments on the manuscript, and the laboratory and institute members for support and discussion throughout this project. Stocks obtained from the Bloomington Drosophila Stock Center (NIH P40OD018537) were used in this study, and the RNAi lines were made by the TRiP project (Office of the Director R24 OD030002: "TRiP resources for modelling human disease", PI: N. Perrimon). Stocks from the Vienna Drosophila Resource Center were also used. The work was funded in part by grants from the Biotechnology and Biological

Sciences Research Council UK (BB/R014507/1, BB/S014357/1, BB/W013525/1, and BB/Y000919/1) to N.A. and Bangabandhu Overseas Scholarship University of Dhaka to A.F.S.

## Author contributions

Conceptualization: L.F. and N.A. Methodology: A.F.S., D.S., S.A., L.F., N.A. Investigation: A.F.S., B.W., I.Ö., D.S., S.A. Visualisation: A.F.S. and N.A. Funding acquisition: A.F.S. and N.A. Project administration: A.F.S. and N.A. Supervision: S.A., L.F., and N.A. Writing – original draft: A.F.S. and N.A. Writing – review & editing: A.F.S., D.S., S.A., L.F., and N.A.

## Competing interests

The authors declare no competing interests.
