## [Transparent Peer Review file · Nature Communications]

β -adrenergic-like signalling engages *CrebB* in *Drosophila* gut to promote female longevity

Corresponding Author: Professor Nazif Alic

Version 0:

Reviewer comments:

Reviewer #1

(Remarks to the Author)

The authors investigate how the invertebrate adrenergic-like signalling system influences ageing and longevity, using *Drosophila melanogaster* as a model organism. By manipulating tyramine (TA) and octopamine (OA) signalling—to draw the parallel with the vertebrate adrenergic hormones—they explore their impact on physiology and lifespan. They find that increased neuronal synthesis of TA extends lifespan and improves neuromuscular and digestive function, particularly in females, while OA had marginal benefits in males. The study further shows that gut-specific activation of the β -adrenergic-like signalling pathway, mediated by *CrebB*, can delay ageing. This body of work reveals how localised, i.e. in a specific tissue, adrenergic-like signalling can regulate animal longevity, offering new insights into neuroendocrine control of ageing. The study provides new insights on how a very important endocrine axis, contributes to healthy ageing. The results are exciting especially in the context of sexual dimorphism. I would be happy to support this effort if the authors address some specific concerns listed below.

TA and OA are produced by the sequential action of tyrosine decarboxylase (Tdc) and tyramine b hydroxylase (Tbh). The authors, use constitutive overexpression of Tdc within its native domain of expression, this leads to 20% lifespan extension.

1. Is there a specific reason why this experiment is done constitutively? In this case the authors cannot rule out developmental priming of these flies for longer lifespan. Thus, the answer of this experiment is not what is the adrenergic signalling good for in the context of an animal that is ageing, but a rather in animal that has highly active of adrenergic signalling their entire life.
2. It is important to show where exactly is the *tdc2*-driver line expressed in the tissues relevant to the experiments the authors describe; they elute to neurons. These are general drivers often expressed in more tissues, relevant to this body of work, neuronal drivers very often are expressed in the gut epithelium.
3. In Supplementary figure 01 it seems that the UAS vs Gal4 controls are significantly different in terms of weight but not in the feeding can the authors comment on the statistics and why is that so?
4. In the case of the *Tdc2>Tdc2* flies, the authors comment that the flies are smaller, do they have insulin signalling defects?
5. The authors show PER, as a proxy for feeding however, I would highly recommend they use a more direct way of measuring feeding, since PER only shows willingness to ingest.
6. The *Tdc2* overexpressing flies, showed reduced locomotion overall. Is the lifespan benefit due to this reduced locomotion over lifetime and could this effect be independent of TA per se?
7. When manipulating *Tdc2* in young age, in Figure 1E, it looks like there is a locomotion worsening phenotype in both sexes and this gets reversed from 35 days old flies and then flat at 50 days. So how does that work?
8. Like my point (02) the authors should show the domain of expression of *Tbh-Gal4* line used in the nervous system and the intestine at least.
9. When feeding OA, males show a detectable increase, so is OA metabolised differently or not absorbed? what is the nature of the sex bias? Are for instance *tbh* levels sexually dimorphic?
10. It would be nice to see what happens in the TA-fed adult females in terms of smurfing and locomotion, to better make sense of the constitutive results shown earlier.
11. Like in point 2, please show where TIGS is expressed in which cell types of the gut epithelium and is it restricted to it?
12. The authors mention that *Octb2R* manipulations in the fat body failed to “significantly impact” longevity, does that mean that they had a minor effect? Can the authors clarify what drivers were used for this experiment and where they are expressed?

13. The authors come to the conclusion that the gut is the important site of activation of b-adrenergic signalling to extend lifespan, it thus becomes important that they clarify the cell-types within the gut epithelium that they have manipulated in their experiments and very interesting to narrow down a time-window in which manipulating this pathway in those cells is critical for lifespan extension.

Minor concerns

1. The authors should try make figure legends clear as to what experiment is done. Neuronal vs TDC overexpression in the neuronal manipulation what driver was used, is it simply TDC?
2. The authors note that Tbh>Tbh manipulations in males have "small but significant effect" what is the %, place it in the text or in the figure legend.

Reviewer #2

(Remarks to the Author)

Firstly, in conclusion, the reviewer believes that the manuscript by Sumit et al. does not meet the high standards required for publication in Nature Communications.

The authors' primary conclusion, drawn from their experiments with *Drosophila melanogaster*, is that the monoamine neurotransmitter tyramine exerts an anti-ageing effect through the gut-expressed tyramine receptor Octb2R and its downstream pathways involving PKA and CreB. While the authors present extensive data in the manuscript, the core evidence supporting this conclusion can be summarised as follows:

1. Neuronal overexpression of Tdc2, which is involved in the biosynthesis of tyramine from tyrosine, has an anti-ageing effect on lifespan, motility, and intestinal barrier function in both male and female flies.
2. Similar anti-ageing effects are observed when tyramine is fed to adult flies.
3. Overexpression of Octb2R, activated PKA, or CreB also extends lifespan.

The reviewer acknowledges the scientific significance of investigating the relationship between monoamines and lifespan and finds the anti-ageing effect of tyramine intriguing. However, the data presented in this study are insufficient to support the bold conclusions drawn. The critical shortcoming lies in the minimal exploration of lifespan or ageing phenotypes when tyramine or its downstream signalling pathways are suppressed (with the sole exception of RNAi targeting CreB). The absence of experiments using loss-of-function models for Tdc2, Octb2R, or PKA is a glaring omission for drawing robust conclusions. Furthermore, in experiments involving tyramine feeding, it is essential to test whether the observed effects are abolished by the suppression of Octb2R, PKA, or CreB. Without such foundational genetic analyses to establish hierarchical relationships, the reviewer must conclude that the authors' assertions regarding the connection between monoamines and ageing are premature.

Additional Concerns:

I) Introduction: The paper opens with the intriguing topic of the "fight or flight" response in animal physiology and behaviour, yet this study solely focuses on monoamines and does not address "fight or flight" at all. The introductory sentence should be significantly revised to align with the study's content.

II) Feeding assays: The feeding assays described on page 5 are inadequate as they rely solely on proboscis extension rather than actual feeding. Incorporating experiments using more standardised feeding measurement methods, such as the CAFE assay, would strengthen the study.

III) Page 8: The authors state: "To examine the effects of increased TA and OA solely in adulthood, we administered the two compounds orally from day two of adulthood...". To specifically examine adult phenotypes, the authors could employ a system that manipulates gene expression exclusively in adults by utilising temperature-sensitive GAL80.

IV) Lifespan discrepancy: While Tdc2 overexpression extends the lifespan of both males and females, tyramine feeding only extends the lifespan of females. How do the authors account for this discrepancy in longevity phenotypes under these experimental conditions? There appears to be no clear discussion in the manuscript.

V) Figure 3B legend (Page 11): The doses of TA and OA used in the experiments should be explicitly stated in the figure legend to aid the readers' understanding.

VI) Page 12, second paragraph: The authors state that the amount of OA increases in both males and females fed 25 µg/ml OA. However, according to Figure 3B, this does not appear to be the case. The reviewer observes no change in the amount of OA in males.

VII) Page 15, line 1: The suggestion that the effect of TA might be paracrine or endocrine is interesting. If so, does the amount of TA in the hemolymph (rather than the whole-body extract) change with age? Additionally, does Tdc2 knockdown reduce TA levels in the hemolymph, and does Tdc2 overexpression increase them?

VIII) Fly stocks and husbandry (Page 21): For *Drosophila* strains obtained from stock centres such as BDSC, specifying the lineage numbers would be helpful for readers.

Reviewer #3

(Remarks to the Author)

In this study, Sumit et al. demonstrated that increasing TA and OA levels could promote longevity in *Drosophila* in a sexual dimorphic manner, likely through the PKA/CREB cascade in the gut. However, the data is currently preliminary and insufficient to support these conclusions.

Key points:

1. The authors primarily altered TA or OA levels through genetic misexpression or dietary supplementation. Do these monoamine levels change during the ageing process?

Since Tdc-expressing bacteria in the gut microbiota is a rich source of TA and/or OA (Ma 2024, EMBO), the conclusion that the activation of the PKA/CREB pathway by neuronal TA/OA may be overestimated. At the very least, the authors should have conducted experiments to verify these conclusions in axenic or germ-free conditions.

2. The authors of the study found that the expression of active PKA or CREB in the gut was specific to the anti-ageing effect of TA. Furthermore, RNAseq analysis indicated that the DEGs were comparable between the TA and CREB overexpressing groups. Consequently, they concluded that TA exerts its effect on lifespan extension through the PKA/CREB cascade. However, it should be noted that these correlative data is insufficient to provide robust support for this conclusion.

At the very least, the authors should have tested whether TA supplementation could still extend lifespan when PKA/CREB was silenced in the gut. Meanwhile, the authors only examined systemic CREB activity by CRE-LUC. It would be critical to examine whether CREB activity (by CRE-luc or antibody staining) or PKA activity (by cAMP kit, Epacs etc) in gut was altered by TA feeding.

3. The author is likely to present a model that the neuronal TA promotes gut PKA/CREB activity through the brain-gut axis. If it is the case, the authors should manipulate TA level in the brain while check the enteric PKA/CREB activity simultaneously. LexA combined with Gal4 could be used to address this question.

4. The authors found TA supplementation in the food can extend lifespan in a dosage dependent manner with the longest extension observed on 100ug/ml. Is 100ug/ml close to the physiological level of TA?

TA is nearly insoluble in water. As stated by the authors in the Method section, the TA used in this study was dissolved in ddH₂O. It is imperative to measure the enteric TA level in the gut when TA is supplemented at varying concentrations.

5. A related question: the authors mentioned that TA feeding didn't change the feeding behavior by examining the proboscis extension, which only assess the feeding frequency. The authors should also compare the food intake by CAFÉ assay.

6. Does Tdc2GAL4 also expressed in the gut? What about TIGSGal4? This is of particular relevance given that the authors' research focuses on the distal regulation of TA in the gut. It is imperative that the specificity of these drivers is carefully verified.

Version 1:

Reviewer comments:

Reviewer #2

(Remarks to the Author)

This reviewer acknowledges that the authors have made substantial efforts to address many of the points and concerns raised by me and the other reviewers in the revised manuscript. It is particularly commendable that the authors were able to detect TA in the hemolymph, as shown in Figure 6B, thereby providing support for the hypothesis that TA may function in an endocrine manner.

The reviewer's only remaining concern is that Figure 7 does not include data showing the loss of TA's effect in TIGS>Oct β 3R-RNAi (RNAi knockdown), in addition to TIGS>Creb-RNAi. The reviewer still believe that focusing on the loss-of-function of the octopamine receptor—not just Creb—is still crucial for a proper understanding of the direct effect of TA on the gut. That said, the reviewer will defer to the Editor's judgment as to whether these additional data should be required.

Reviewer #3

(Remarks to the Author)

The authors have addressed most of my concerns, and the manuscript is significantly improved. However, point 3 was inadequately addressed.

Throughout the manuscript, the authors rely on global genetic overexpression strategies (Tdc2-Gal4 > UAS-Tdc2 in Figures 1-2; Tbh-Gal4 > UAS-Tbh in Figure 3) and systemic feeding experiments (Figure 4) to manipulate tyramine (TA) levels. However, these approaches fail to resolve fundamental questions regarding TA's cellular origin, spatiotemporal regulation during aging, and the basis for observed sexual dimorphism.

Critically, the source(s) of physiologically relevant TA pools remain unidentified. Potential contributors—including specific neuronal subtypes (CNS or enteric), enteroendocrine cells, or gut microbiota—were not distinguished experimentally. To establish mechanistic causality, cell-type-specific manipulation (e.g., intersectional genetics) and temporal control of TA

synthesis are essential. Furthermore, the work does not address how TA production is dynamically regulated throughout aging or how sex-specific differences arise.

The current model lacks spatial and temporal resolution, need more experimental evidence in addition to more discussion.

3, The author is likely to present a model that the neuronal TA promotes gut PKA/CREB activity through the brain-gut axis. If it is the case, the authors should manipulate TA level in the brain while check the enteric PKA/CREB activity simultaneously. LexA combined with Gal4 could be used to address this question.

We are not presenting a model including a brain-gut axis. The key conclusion of our manuscript is that the activation of b-adrenergic-like signalling can promote longevity. Our data are consistent with either neuronal, endocrine/paracrine or microbial source of TA that acts on the signalling pathway within the gut. We have now made this clear in the discussion.

Version 2:

Reviewer comments:

Reviewer #2

(Remarks to the Author)

The reviewer has no additional comments regarding the manuscript.

We thank the reviewers for their assessment of our manuscript; their reviews were an invaluable contribution to our work and significantly helped us improve the manuscript. We have now performed substantial additional experiments, adding over 20 new figure panels to the revised manuscript, including over 10 additional ageing experiments. We have also edited the manuscript text as requested by the reviewers. Any substantial changes are marked in red in the manuscript file. Detailed replies to reviewer comments are given below.

Reviewer #1 (Remarks to the Author):

The authors investigate how the invertebrate adrenergic-like signalling system influences ageing and longevity, using *Drosophila melanogaster* as a model organism. By manipulating tyramine (TA) and octopamine (OA) signalling—to draw the parallel with the vertebrate adrenergic hormones—they explore their impact on physiology and lifespan. They find that increased neuronal synthesis of TA extends lifespan and improves neuromuscular and digestive function, particularly in females, while OA had marginal benefits in males. The study further shows that gut-specific activation of the β -adrenergic-like signalling pathway, mediated by CrebB, can delay ageing. This body of work reveals how localised, i.e. in a specific tissue, adrenergic-like signalling can regulate animal longevity, offering new insights into neuroendocrine control of ageing.

The study provides new insights on how a very important endocrine axis, contributes to healthy ageing. The results are exciting especially in the context of sexual dimorphism. I would be happy to support this effort if the authors address some specific concerns listed below.

We thank the reviewer for appreciating that our study provides new insights into the role of a very important endocrine axis in animal ageing. We detail below how we have addressed the concerns raised.

TA and OA are produced by the sequential action of tyrosine decarboxylase (Tdc) and tyramine b hydroxylase (Tbh) The authors, use constitutive overexpression of Tdc within its native domain of expression, this leads to 20% lifespan extension.

1. Is there a specific reason why this experiment is done constitutively? In this case the authors cannot rule out developmental priming of these flies for longer lifespan. Thus, the answer of this experiment is not what is the adrenergic signalling good for in the context of an animal that is ageing, but a rather in animal that has highly active of adrenergic signalling their entire life.

We used the constitutive expression of *Tdc2* as we wanted to increase its expression only in cells in which it is naturally expressed and the only suitable driver is a constitutive *GAL4* driver (*Tdc2-GAL4*). We agree that the longevity phenotype in *Tdc2>Tdc2* may be caused in part by the developmental effects and this is the reason we went on to employ two further models: adult-onset feeding of TA (and OA) and adult-onset manipulation of the intracellular pathway components in the gut. The lifespan extension we observed in these cases can be ascribed to the functioning of the pathway in adults. We have now made adjustments to the text to better emphasise this point.

2. It is important to show where exactly is the *tdc2*-driver line expressed in the tissues relevant to the experiments the authors describe; they elute to neurons. These are general drivers often expressed in more tissues, relevant to this body of work, neuronal drivers very often are expressed in the gut epithelium.

We thank the reviewer for this comment: we realised that we have not adequately described the drivers used in our work but merely included the relevant references in the text. *Tdc2* expression has been characterised in detail and it is restricted to neurons (Cole et al., 2005). To recapitulate this expression, a driver was created using a fragment of the *Tdc2* gene (the promoter region). This is the *Tdc2-GAL4* we used, and its expression was carefully characterised (Cole et al., 2005). We have now added these details in the text. The driver has been used extensively, with 137 publications reported in FlyBase (<https://flybase.org/reports/FBtp0127561>); we are not aware of any researcher reporting non-neuronal expression. However, we do agree that establishing if any expression exists in the gut is important for our work. For this reason, we have examined the expression of the driver in female brain and gut and confirmed brain expression with no expression observed in the gut (Supplementary Figure 7).

3. In Supplementary figure 01 it seems that the UAS vs Gal4 controls are significantly different in terms of weight but not in the feeding can the authors comment on the statistics and why is that so?

The differences between the two controls can occur due to insertional mutagenesis (each is inserted somewhere in the genome), effects of driver (simply due to expression of *GAL4*), some leakiness of the transgene (low expression in the absence of *GAL4*), or differences in background (simply due to sampling from the outbred population during backcrossing; or due to genetic drift). This is the reason why we used *a priori* contrasts in our analyses that allow robust statistical testing for significant differences between the test genotype and both controls. We have now explained this in more detail in the text at the start of the results section, not referring to any specific figure or phenotype but as a general comment. Please note, with respect to adult feeding and adult body size: body weight is mostly the result of growth during development and may not correlate to adult feeding.

4. In the case of the *Tdc2>Tdc2* flies, the authors comment that the flies are smaller, do they have insulin signalling defects?

Indeed, the reduced size, increased lifespan and reduced fecundity in females are similar to the effect of mutations in the insulin-like signalling pathway. We have now directly tested for alterations in insulin-like signalling by looking at the phosphorylation of the two kinases downstream of the Insulin Receptor: ERK and AKT, specifically in females as this signalling pathway is better characterised in this sex. We did not observe a reduction in the levels of phosphorylated ERK (pERK) or phosphorylated AKT, indicating that insulin signalling is not inhibited (Figure 1E and F, Supplementary Figure 1K and Supplementary Figure 10). We actually observe a slight increase in pERK (Figure 1F, Supplementary Figure 1K), and this may be consistent with the egg-retention suggested to occur upon increased TA levels (Cole et al., 2005).

5. The authors show PER, as a proxy for feeding however, I would highly recommend they use a more direct way of measuring feeding, since PER only shows willingness to ingest.

We thank the reviewer for this suggestion. We have now quantified the amount of food ingested, using a blue food dye consumption assay as has been done by others (Wong et al., 2009) and which can be applied directly under the same conditions that we use for other phenotypes, specifically solid food in a vial. This confirms our conclusions based on proboscis extension assay (PER): there is reduced feeding in *Tdc2>Tdc2* females but not males (Supplementary Figure 1I). We also confirmed no effect on feeding from oral TA administration (Supplementary Figure 4G) or from gut induction of *Octβ2R* (Supplementary Figure 5D).

6. The *Tdc2* overexpressing flies, showed reduced locomotion overall. Is the lifespan benefit due to this reduced locomotion over lifetime and could this effect be independent of TA per se?

This is a very interesting comment. It is possible that the two phenotypes are linked and we have now mentioned this possibility in the discussion. Our new data looking at climbing in females fed TA (Figure 4D) indicate that their ability to climb is only marginally (Figure 4D, $p=0.09$, Supplementary Figure 4H) affected by TA but that TA causes a delay in the age-related decline of climbing ability (Figure 4D, $p=0.036$, Supplementary Figure 4H). TA-fed females have an extended lifespan. This may indicate that the link between the two (activity and lifespan) is not complete, and furthermore that the native effect on locomotion is not independent of TA. We have not investigated this further as we have identified at least one key longevity mechanism: β -adrenergic like signalling in the female gut. We hope to identify additional mechanisms in our future work.

7. When manipulating *Tdc2* in young age, in Figure 1E, it looks like there is a locomotion worsening phenotype in both sexes and this gets reversed from 35 days old flies and then flat at 50 days. So how does that work?

Tdc2>Tdc2 flies have a poorer climbing ability than the two controls at young age. With ageing, the climbing ability of the controls declines faster than the climbing ability of the *Tdc2>Tdc2* flies so that at later ages *Tdc2>Tdc2* flies climb better than controls. This is confirmed with our statistical analysis: the age-related decline in climbing ability

is slower (delayed) in *Tdc2>Tdc2* flies; they are less affected by ageing. We have added a more careful description of the data and the statistical analysis in the text.

8. Like my point (02) the authors should show the domain of expression of *Tbh-Gal4* line used in the nervous system and the intestine at least.

We have now provided more detailed information on the drivers used in our study, including *Tbh-GAL4*, in the text. Similarly to *Tdc2*, the expression of *Tbh* has been characterised in detail: it is restricted to specific populations of neurons and not in any non-neuronal tissues (Monastirioti, 2003). The driver we use was generated using a genomic fragment of the *Tbh* gene to drive *GAL4* at the Janelia Research Campus; its expression in the adult brain has been characterised and is available in the supplement of the publication of the driver collection (Jenett et al., 2012). We also note that this driver is unlikely to recapitulate the full expression pattern of *Tbh*, as it is one of several drivers generated from *Tbh* enhancers which all appear to drive in subsets of octopaminergic neurons [please see (Iliadi et al., 2017)]. We now highlight the caveats of using this driver. Please note that: 1) the effect we see on lifespan in males is confirmed by OA feeding, independently of using this driver; 2) this driver is not important for our main conclusion, that β -adrenergic-like signalling boosts longevity.

9. When feeding OA, males show a detectable increase, so is OA metabolised differently or not absorbed? what is the nature of the sex bias? Are for instance *tbh* levels sexually dimorphic?

We thank the reviewer for raising this point. Sexual dimorphism within the OA/TA system in flies is increasingly appreciated and we have now extended our consideration of this question in the discussion section of the manuscript, providing some answers to the questions posed by the reviewer. There is known sexual dimorphism in *Tbh* expression, for example some of the *Tbh* expressing neurons are sexually dimorphic (Rezaval et al., 2014), and OA mediates adaptations to exercise specifically in males (Sujkowski et al., 2017). In the case of longevity-promoting effects of TA, we think the dimorphism is linked to the known dimorphism in gut physiology and its impact on ageing, and this is now included in the discussion section of the

manuscript. As for OA, we did not focus on the effect of OA on male longevity, since the effect we observed is quite minor. For this reason, we do not think we are in a position to propose a mechanism for the dimorphism but we hope to focus on this in subsequent work.

10. It would be nice to see what happens in the TA-fed adult females in terms of smurfing and locomotion, to better make sense of the constitutive results shown earlier.

We have now assessed age-related changes in climbing and gut barrier function in females fed TA. We find that TA significantly delays the age-related decline in climbing ability (Figure 4D, Supplementary Figure 4H) and significantly reduces the occurrence of gut-barrier failure (Figure 4E and Supplementary Figure 4I), similarly to the observations in *Tdc2>Tdc2* flies, indicating that adult onset increase in TA is sufficient to achieve these benefits.

11. Like in point 2, please show where TIGS is expressed in which cell types of the gut epithelium and is it restricted to it?

This driver has been extensively characterised by us and others. it allows for expression restricted to the gut upon feeding of the RU486 inducer (Poirier et al., 2008). Within the gut, the expression can be observed in both enterocytes and the intestinal stem cells (Alic et al., 2014; Filer et al., 2017). We have now noted this in the text.

12. The authors mention that Octb2R manipulations in the fat body failed to “significantly impact” longevity, does that mean that they had a minor effect? Can the authors clarify what drivers were used for this experiment and where they are expressed?

We apologise that this was unclear. We meant that there was no effect observed – neither negative nor positive. This experiment was performed with another extensively characterised driver, *S106*, which drives in the abdominal fat body. We have now added details in the text.

13. The authors come to the conclusion that the gut is the important site of activation of adrenergic signalling to extend lifespan, it thus becomes important that they clarify the cell types within the gut epithelium that they have manipulated in their experiments and very interesting to narrow down a time-window in which manipulating this pathway in those cells is critical for lifespan extension.

We thank the reviewer for this comment. In addition to detailing the cell types in which the TIGS driver is active (enterocytes and intestinal stem cells) we have now directly manipulated components of β -adrenergic-like signalling pathway with additional, cell type-restricted, inducible drivers. We now show that induction of the activated form of PKA or CrebB in enterocytes can extend female lifespan, while the induction of the same construct in the intestinal stem cells does not (Figure 5D and E, Supplementary Figure 6H; Figure 7C, Supplementary Figure 8H-J). For these experiments, we used previously characterised drivers *Mex1GS*, *GS5966* and *GS5961* (Biteau et al., 2010; Mathur et al., 2010; Soule et al., 2020). We also clarify that all the inducible drivers were activated from day two of adulthood, highlighting that the manipulation of the pathway in adulthood is sufficient for longevity.

Minor concerns

1. The authors should try make figure legends clear as to what experiment is done. Neuronal vs TDC overexpression in the neuronal manipulation what driver was used, is it simply TDC?

We have added information to figure captions to clarify this.

2. The authors note that *Tbh>Tbh* manipulations in males have “small but significant effect” what is the %, place it in the text or in the figure legend.

We have now added this information to the text: there was no increase in median lifespan but a 5% increase in maximum lifespan (oldest 5%).

Reviewer #2 (Remarks to the Author):

Firstly, in conclusion, the reviewer believes that the manuscript by Sumit et al. does not meet the high standards required for publication in Nature Communications. The authors' primary conclusion, drawn from their experiments with *Drosophila melanogaster*, is that the monoamine neurotransmitter tyramine exerts an anti-ageing effect through the gut-expressed tyramine receptor Octb2R and its downstream pathways involving PKA and CreB. While the authors present extensive data in the manuscript, the core evidence supporting this conclusion can be summarised as follows:

1. Neuronal overexpression of Tdc2, which is involved in the biosynthesis of tyramine from tyrosine, has an anti-ageing effect on lifespan, motility, and intestinal barrier function in both male and female flies.
2. Similar anti-ageing effects are observed when tyramine is fed to adult flies.
3. Overexpression of Octb2R, activated PKA, or CreB also extends lifespan.

The reviewer acknowledges the scientific significance of investigating the relationship between monoamines and lifespan and finds the anti-ageing effect of tyramine intriguing. However, the data presented in this study are insufficient to support the bold conclusions drawn. The critical shortcoming lies in the minimal exploration of lifespan or ageing phenotypes when tyramine or its downstream signalling pathways are suppressed (with the sole exception of RNAi targeting CreB). The absence of experiments using loss-of-function models for Tdc2, Octb2R, or PKA is a glaring omission for drawing robust conclusions.

We are pleased that the reviewer acknowledges the scientific significance of our investigation and we thank the reviewer for the comments. The reviewer believes that experiments on loss-of-function models are essential for robust conclusions. We believe this perception may have arisen from our insufficient explanation of the experimental approaches employed in the ageing field. In the research on the biology of ageing, it is well established that the most informative interventions are those that result in an extension of lifespan, as they directly demonstrate what limits lifespan in the wild type. Indeed, all important publications in the field are based on observations of lifespan extension, from the initial characterisation of the longevity benefits of reduced insulin-like signalling (Kenyon et al., 1993) to the identification of the TORC1

pathway as an evolutionarily conserved driver of ageing (Bjedov et al., 2010; Harrison et al., 2009; Vellai et al., 2003). In contrast, the interventions that shorten lifespan of the wild type are not necessarily informative about ageing as there are numerous ways to make an animal sick, most of which have nothing to do with ageing.

Our study started with a hypothesis that increased TA/OA would be beneficial for longevity. This led us to show that increasing TA, either genetically or by oral administration, or increasing β -adrenergic-like signalling in the gut, can extend lifespan in females. Hence, the experiments that we performed, as they result in an extension of lifespan in an otherwise wild-type, outbred and healthy fly stock, are both necessary and sufficient to draw robust conclusions about the ageing process. We have now explained this in more detail at the start of the results section.

Furthermore, in experiments involving tyramine feeding, it is essential to test whether the observed effects are abolished by the suppression of Octb2R, PKA, or CreB. Without such foundational genetic analyses to establish hierarchical relationships, the reviewer must conclude that the authors' assertions regarding the connection between monoamines and ageing are premature.

We agree that interaction (epistasis) experiments are crucial for establishing hierarchical relationships within the pathway. We show that the longevity effect of TA feeding is abolished by knockdown of *CrebB* in the adult female gut (Figure 7A, Supplementary Figure 8E). We believe this satisfies the reviewer's request. But we didn't stop there: we performed two additional epistasis experiments.

In the ageing field there is a preference to test whether manipulations that are individually beneficial show additive effects when combined as a means of determining if they act within the same longevity pathway as this avoids the use of sick or unhealthy mutants; non-additive effects indicate the interventions act within the same longevity pathway. This approach has been used in numerous publications (Castillo-Quan et al., 2019; Filer et al., 2017; Gkioni et al., 2025; Urena et al., 2024). We did this here by looking at the combined effects of TA feeding and gut-restricted induction of *Oct β 2R*. We found the two were not additive indicating they act in the same pathway (Figure 7E, Supplementary Figure 9C). Furthermore, we also combined the gut-restricted

expression of *Octβ2R* and *CrebB* and found their effects non-additive (Figure 7D, Supplementary Figure 9B), concluding they act in the same pathway.

These three interaction (epistasis) experiments were carefully statistically analysed using the appropriate Cox Proportional Hazard models (Supplementary Figure 8E, 9B and C). All three point to the same conclusion. This conclusion is further reinforced with additional, new observations of *CrebB* activation in the gut upon TA feeding (Figure 6G). Therefore, we believe that the main conclusion of our work, that activation of β-adrenergic-like signalling promotes longevity, is robust and substantiated.

Additional Concerns:

I) Introduction: The paper opens with the intriguing topic of the "fight or flight" response in animal physiology and behaviour, yet this study solely focuses on monoamines and does not address "fight or flight" at all. The introductory sentence should be significantly revised to align with the study's content.

We have edited the abstract and introduction accordingly.

II) Feeding assays: The feeding assays described on page 5 are inadequate as they rely solely on proboscis extension rather than actual feeding. Incorporating experiments using more standardised feeding measurement methods, such as the CAFE assay, would strengthen the study.

We have now quantified the amount of food consumed by the flies, adapting a described food consumption assay (Wong et al., 2009). This confirms the initial findings with the proboscis extension assay: there is reduced feeding in *Tdc2>Tdc2* females (Supplementary Figure 1I) but not males (Supplementary Figure 1I), there is no effect on feeding from oral TA administration (Supplementary Figure 4G) or from gut induction of *Octβ2R* (Supplementary Figure 5D).

III) Page 8: The authors state: "To examine the effects of increased TA and OA solely in adulthood, we administered the two compounds orally from day two of adulthood...". To specifically examine adult phenotypes, the authors could employ a system that

manipulates gene expression exclusively in adults by utilising temperature-sensitive GAL80.

Temperature-sensitive *GAL80* is not ideal for lifespan assays as temperature massively alters the lifespan of fruit flies, with effects that can persist (Mair et al., 2003). For this reason, we do not use this system. Rather, we have employed the more appropriate, inducible, GeneSwitch system, in over 20 individual lifespan experiments to support our main conclusions: that adult-onset activation of β -adrenergic like signalling promotes longevity. This inducible system is extensively used in fly ageing studies.

IV) Lifespan discrepancy: While *Tdc2* overexpression extends the lifespan of both males and females, tyramine feeding only extends the lifespan of females. How do the authors account for this discrepancy in longevity phenotypes under these experimental conditions? There appears to be no clear discussion in the manuscript.

This difference (not a discrepancy) likely results from differences in TA administration. For example, it may be due to specific effect of TA directly at sites innervated by tyraminerpic neurons in males that cannot be recapitulated by TA feeding; similarly, some deficits in neuronal OA production cannot be rescued by OA feeding (Iliadi et al., 2017). We have added this comment to the manuscript. We believe this is part of the sexual dimorphism we report e.g. TA causes sustained *CrebB* activation in females but not in males. We have also extended our consideration of this sexual dimorphism in the discussion section of the manuscript.

V) Figure 3B legend (Page 11): The doses of TA and OA used in the experiments should be explicitly stated in the figure legend to aid the readers' understanding.

We have added this information directly to the main figures, for clarity.

VI) Page 12, second paragraph: The authors state that the amount of OA increases in both males and females fed 25 $\mu\text{g/ml}$ OA. However, according to Figure 3B, this does

not appear to be the case. The reviewer observes no change in the amount of OA in males.

We thank the reviewer for this comment. The statistical model indicated that the amount of OA is increased ($p=0.018$) and that there is no significant difference between the sexes in this effect (interaction $p=0.12$, Supplementary Figure 4B). This is what was meant by this statement. The average OA levels in males increase from 30.2 to 32.6 nmols/g, which we agree appears less than the increase observed in females (Figure 4B). We have now qualified this statement.

VII) Page 15, line 1: The suggestion that the effect of TA might be paracrine or endocrine is interesting. If so, does the amount of TA in the haemolymph (rather than the whole-body extract) change with age? Additionally, does Tdc2 knockdown reduce TA levels in the haemolymph, and does Tdc2 overexpression increase them?

We thank the reviewer for this comment as it prompted us to examine if TA is present in adult haemolymph; we are not aware of anybody reporting this previously. Using LC-MS/MS we have now detected TA in haemolymph obtained from adult females (Figure 6A), which is consistent with a potential endocrine function of TA. Furthermore, we show that haemolymph TA level is increased when flies are fed TA (Figure 6B).

VIII) Fly stocks and husbandry (Page 21): For *Drosophila* strains obtained from stock centres such as BDSC, specifying the lineage numbers would be helpful for readers.

We have added this information.

Reviewer #3 (Remarks to the Author):

In this study, Sumit et al. demonstrated that increasing TA and OA levels could promote longevity in *Drosophila* in a sexual dimorphic manner, likely through the PKA/CREB cascade in the gut. However, the data is currently preliminary and insufficient to support these conclusions.

We thank the reviewer for the assessment of our work. We have performed an extensive characterisation of the impact of TA/OA signalling on longevity, including

over 40 lifespan experiments, detailed characterisation of health, OA/TA levels, transcription factor activity and gene expression, using several models of altered TA/OA signalling. We disagree that this is preliminary. Still, we have taken the comments by the reviewer seriously and performed additional experimental work as detailed below.

Key points:

1. The authors primarily altered TA or OA levels through genetic misexpression or dietary supplementation. Do these monoamine levels change during the ageing process? Since Tdc-expressing bacteria in the gut microbiota is a rich source of TA and/or OA (Ma 2024, EMBO), the conclusion that the activation of the PKA/CREB pathway by neuronal TA/OA may be overestimated. At the very least, the authors should have conducted experiments to verify these conclusions in axenic or germ-free conditions.

We thank the reviewer for the comments and provide several points in reply. (1) We now summarise what is known about changes in OA/TA levels across a fly's lifetime at the start of the results. (2) We thank the reviewer for pointing out the interesting paper from the Dang lab. We have now included the discussion of this work in our manuscript. The work from the Dang lab focuses on what happens under condition when flies are fed a diet rich in fat, whereas our flies are fed a healthy diet, an important difference between the two studies. (3) However, we do not dispute that the microbiota may contribute to TA levels in flies and have now noted this in the discussion section of the manuscript. As we now explicitly state in the discussion, the source of TA may be neuronal, from non-neuronal tissues or from the microbiota; this does not affect our main conclusion. (4) Importantly, the main conclusion of our paper is that increased β -adrenergic-like signalling can promote longevity, and several experiments supporting this conclusion are not based on TA/OA manipulations, but rather the direct manipulation of the components of the β -adrenergic-like signalling pathway in the adult gut, using inducible drivers. (5) Lastly, several studies have reported an extension of lifespan due to removal of the microbiota [e.g. (Clark et al., 2015)], which complicates the interpretation of any axenic or germ-free experiment. Because of all this, we do

not see the relevance of conducting these experiments in germ-free or axenic conditions.

2, The authors of the study found that the expression of active PKA or CREB in the gut was specific to the anti-ageing effect of TA. Furthermore, RNAseq analysis indicated that the DEGs were comparable between the TA and CREB overexpressing groups. Consequently, they concluded that TA exerts its effect on lifespan extension through the PKA/CREB cascade. However, it should be noted that these correlative data is insufficient to provide robust support for this conclusion.

At the very least, the authors should have tested whether TA supplementation could still extend lifespan when PKA/CREB was silenced in the gut. Meanwhile, the authors only examined systemic CREB activity by CRE-LUC. It would be critical to examine whether CREB activity (by CRE-luc or antibody staining) or PKA activity (by cAMP kit, Epacs etc) in gut was altered by TA feeding.

Indeed, we showed that the effects of TA feeding are abolished by knockdown of *CrebB* in the adult female gut (Figure 7A, Supplementary Figure 8E). This is one of the three interaction experiments that we performed (see reply to reviewer 2 comments; Figure 7D and E, Supplementary Figure 8E and 9B and C). All provide evidence that TA acts in the same pathway as *Octβ2R* and *CrebB* in the gut to impact longevity. We have additionally emphasised this in the manuscript.

We thank the reviewer for prompting us to test whether CRE-luc is induced in the gut by TA feeding. We now report that TA feeding induces the CRE-luc reporter in the female gut (Figure 6G). Additionally, we now show that the set of TA-regulated genes we uncovered overlap significantly with genes responsive to CRTG (Supplementary Figure 8B), a *CrebB* partner, using published RNA-Seq data (Yin et al., 2022).

3, The author is likely to present a model that the neuronal TA promotes gut PKA/CREB activity through the brain-gut axis. If it is the case, the authors should manipulate TA level in the brain while check the enteric PKA/CREB activity simultaneously. LexA combined with Gal4 could be used to address this question.

We are not presenting a model including a brain-gut axis. The key conclusion of our manuscript is that the activation of β -adrenergic-like signalling can promote longevity. Our data are consistent with either neuronal, endocrine/paracrine or microbial source of TA that acts on the signalling pathway within the gut. We have now made this clear in the discussion.

4, The authors found TA supplementation in the food can extend lifespan in a dosage dependent manner with the longest extension observed on 100ug/ml. Is 100ug/ml close to the physiological level of TA?

TA is nearly insoluble in water. As stated by the authors in the Method section, the TA used in this study was dissolved in ddH₂O. It is imperative to measure the enteric TA level in the gut when TA is supplemented at varying concentrations.

We agree with the reviewer that the relevant concentration is not what is in the food but rather what is in the flies. For this reason, we have measured the increase in TA after TA feeding and find the amounts of TA increased in flies by approximately 50% (Figure 4A). We have now also observed TA in haemolymph and found that TA feeding leads to approximately 20% increase in haemolymph TA (Figure 6B). These two measurements are sufficient to show the magnitude of increase in physiologically relevant TA levels.

We now explain in detail that TA and OA administration in food is a frequent experimental paradigm in *Drosophila*. TA and OA feeding can reverse several phenotypes associated with impaired ability to produce these bioamines, including behavioural and reproductive phenotypes (Crocker and Sehgal, 2008; Monastirioti et al., 1996 ; Saraswati et al., 2004; Sujkowski et al., 2017), which indicates TA and OA are taken up by the fly and can reach distal organs in an active form.

TA is water soluble to a concentration of 10 mg/ml; we used a stock of 5 mg/ml and had no problems with solubility.

5, A related question: the authors mentioned that TA feeding didn't change the feeding behavior by examining the proboscis extension, which only assess the feeding frequency. The authors should also compare the food intake by CAFÉ assay.

We have now examined food consumption directly, adopting a described method for quantification of consumption under standard culture conditions (Wong et al., 2009), and confirm the results of the proboscis extension assays: there is reduced feeding in *Tdc2>Tdc2* females (Supplementary Figure 1I) but not males (Supplementary Figure 1I), there is no effect on feeding from oral TA administration (Supplementary Figure 4G) or from gut induction of *Octβ2R* (Supplementary Figure 5D).

6, Does *Tdc2GAL4* also expressed in the gut? What about *TIGSGal4*? This is of particular relevance given that the authors' research focuses on the distal regulation of TA in the gut. It is imperative that the specificity of these drivers is carefully verified.

The expression of the *Tdc2-GAL4* we used has been carefully characterised (Cole et al., 2005). We have now added these details in the text. We do agree that establishing if any expression exists in the gut was important for our work. For this reason, we have examined the expression of the driver in female brain and gut and confirmed brain expression with no expression observed in the gut (Supplementary Figure 7). *TiGS* has also been characterised extensively by us and others. Its expression is restricted to the gut (Poirier et al., 2008) and is expressed in enterocytes and intestinal stem cells (Alic et al., 2014; Filer et al., 2017). We now give a detailed account of this in the results section.

References

- Alic, N., Giannakou, M.E., Papatheodorou, I., Hoddinott, M.P., Andrews, T.D., Bolukbasi, E., and Partridge, L. (2014). Interplay of dFOXO and two ETS-family transcription factors determines lifespan in *Drosophila melanogaster*. *PLoS genetics* 10, e1004619.
- Biteau, B., Karpac, J., Supoyo, S., DeGennaro, M., Lehmann, R., and Jasper, H. (2010). Lifespan Extension by Preserving Proliferative Homeostasis in *Drosophila*. *PLoS genetics* 6.
- Bjedov, I., Toivonen, J.M., Kerr, F., Slack, C., Jacobson, J., Foley, A., and Partridge, L. (2010). Mechanisms of life span extension by rapamycin in the fruit fly *Drosophila melanogaster*. *Cell Metab* 11, 35-46.

- Castillo-Quan, J.I., Tain, L.S., Kinghorn, K.J., Li, L., Grönke, S., Hinze, Y., Blackwell, T.K., Bjedov, I., and Partridge, L. (2019). A triple drug combination targeting components of the nutrient-sensing network maximizes longevity. *Proceedings of the National Academy of Sciences* 116, 20817-20819.
- Clark, R.I., Salazar, A., Yamada, R., Fitz-Gibbon, S., Morselli, M., Alcaraz, J., Rana, A., Rera, M., Pellegrini, M., Ja, W.W., *et al.* (2015). Distinct Shifts in Microbiota Composition during *Drosophila* Aging Impair Intestinal Function and Drive Mortality. *Cell Rep* 12, 1656-1667.
- Cole, S.H., Carney, G.E., McClung, C.A., Willard, S.S., Taylor, B.J., and Hirsh, J. (2005). Two functional but noncomplementing *Drosophila* tyrosine decarboxylase genes: distinct roles for neural tyramine and octopamine in female fertility. *J Biol Chem* 280, 14948-14955.
- Crocker, A., and Sehgal, A. (2008). Octopamine regulates sleep in *drosophila* through protein kinase A-dependent mechanisms. *J Neurosci* 28, 9377-9385.
- Filer, D., Thompson, M.A., Takhaveev, V., Dobson, A.J., Kotronaki, I., Green, J.W.M., Heinemann, M., Tullet, J.M.A., and Alic, N. (2017). RNA polymerase III limits longevity downstream of TORC1. *Nature* 552, 263.
- Gkioni, L., Nespital, T., Baghdadi, M., Monzo, C., Bali, J., Nassr, T., Cremer, A.L., Beyer, A., Deelen, J., Backes, H., *et al.* (2025). The geroprotectors trametinib and rapamycin combine additively to extend mouse healthspan and lifespan. *Nat Aging*.
- Harrison, D.E., Strong, R., Sharp, Z.D., Nelson, J.F., Astle, C.M., Flurkey, K., Nadon, N.L., Wilkinson, J.E., Frenkel, K., Carter, C.S., *et al.* (2009). Rapamycin fed late in life extends lifespan in genetically heterogeneous mice. *Nature* 460, 392-395.
- Iliadi, K.G., Iliadi, N., and Boulianne, G.L. (2017). *Drosophila* mutants lacking octopamine exhibit impairment in aversive olfactory associative learning. *Eur J Neurosci* 46, 2080-2087.
- Jenett, A., Rubin, G.M., Ngo, T.T., Shepherd, D., Murphy, C., Dionne, H., Pfeiffer, B.D., Cavallaro, A., Hall, D., Jeter, J., *et al.* (2012). A GAL4-driver line resource for *Drosophila* neurobiology. *Cell Rep* 2, 991-1001.
- Kenyon, C., Chang, J., Gensch, E., Rudner, A., and Tabtiang, R. (1993). A *C. elegans* mutant that lives twice as long as wild type. *Nature* 366, 461-464.
- Mair, W., Goymer, P., Pletcher, S.D., and Partridge, L. (2003). Demography of dietary restriction and death in *Drosophila*. *Science* 301, 1731-1733.

- Mathur, D., Bost, A., Driver, I., and Ohlstein, B. (2010). A transient niche regulates the specification of *Drosophila* intestinal stem cells. *Science* 327, 210-213.
- Monastirioti, M. (2003). Distinct octopamine cell population residing in the CNS abdominal ganglion controls ovulation in *Drosophila melanogaster*. *Dev Biol* 264, 38-49.
- Monastirioti, M., Linn, C.E., Jr., and White, K. (1996). Characterization of *Drosophila* tyramine beta-hydroxylase gene and isolation of mutant flies lacking octopamine. *J Neurosci* 16, 3900-3911.
- Poirier, L., Shane, A., Zheng, J., and Seroude, L. (2008). Characterization of the *Drosophila* gene-switch system in aging studies: a cautionary tale. *Aging Cell* 7, 758-770.
- Rezaval, C., Nojima, T., Neville, M.C., Lin, A.C., and Goodwin, S.F. (2014). Sexually dimorphic octopaminergic neurons modulate female postmating behaviors in *Drosophila*. *Current biology : CB* 24, 725-730.
- Saraswati, S., Fox, L.E., Soll, D.R., and Wu, C.F. (2004). Tyramine and octopamine have opposite effects on the locomotion of *Drosophila* larvae. *J Neurobiol* 58, 425-441.
- Soule, S., Mellottee, L., Arab, A., Chen, C., and Martin, J.R. (2020). Jouvence a small nucleolar RNA required in the gut extends lifespan in *Drosophila*. *Nat Commun* 11, 987.
- Sujkowski, A., Ramesh, D., Brockmann, A., and Wessells, R. (2017). Octopamine Drives Endurance Exercise Adaptations in *Drosophila*. *Cell Rep* 21, 1809-1823.
- Urena, E., Xu, B., Regan, J.C., Atilano, M.L., Minkley, L.J., Filer, D., Lu, Y.X., Bolukbasi, E., Khericha, M., Alic, N., *et al.* (2024). Trametinib ameliorates aging-associated gut pathology in *Drosophila* females by reducing Pol III activity in intestinal stem cells. *Proc Natl Acad Sci U S A* 121, e2311313121.
- Vellai, T., Takacs-Vellai, K., Zhang, Y., Kovacs, A.L., Orosz, L., and Muller, F. (2003). Genetics: influence of TOR kinase on lifespan in *C. elegans*. *Nature* 426, 620.
- Wong, R., Piper, M.D., Wertheim, B., and Partridge, L. (2009). Quantification of food intake in *Drosophila*. *PLoS One* 4, e6063.
- Yin, Y., Ma, P., Wang, S., Zhang, Y., Han, R., Huo, C., Wu, M., and Deng, H. (2022). The CRTC-CREB axis functions as a transcriptional sensor to protect against proteotoxic stress in *Drosophila*. *Cell Death Dis* 13, 688.

We thank the reviewers for their assessment of our manuscript. In this second revision, prompted by the comments of reviewer 2, we have backcrossed an additional transgenic line and performed an additional lifespan experiment. The new data are in line with our overall conclusions.

Changes made to the manuscript are marked in orange in the manuscript file (we have kept the extensive changes made during the initial revision in red). Detailed replies to reviewer comments are given below.

Reviewer #2 (Remarks to the Author):

This reviewer acknowledges that the authors have made substantial efforts to address many of the points and concerns raised by me and the other reviewers in the revised manuscript. It is particularly commendable that the authors were able to detect TA in the hemolymph, as shown in Figure 6B, thereby providing support for the hypothesis that TA may function in an endocrine manner.

The reviewer's only remaining concern is that Figure 7 does not include data showing the loss of TA's effect in TIGS>Oct β 3R-RNAi (RNAi knockdown), in addition to TIGS>Creb-RNAi. The reviewer still believe that focusing on the loss-of-function of the octopamine receptor—not just Creb—is still crucial for a proper understanding of the direct effect of TA on the gut. That said, the reviewer will defer to the Editor's judgment as to whether these additional data should be required.

We thank the reviewer for the positive assessment of our revised manuscript and for the appreciation of all the work that went into addressing the 1st round of reviewer comments. We have now made an additional revision to the manuscript: We have completed the backcrossing of an RNAi line targeting *Oct β 2R* which was used previously in the excellent work from the Wessells group (Sujkowski et al., 2020), and have now tested whether the induction of *Oct β 2R^{RNAi}* in the adult female gut alters the lifespan response to TA. The results are described in the manuscript as follows (lines 365-373):

Lastly, we tested whether knocking down *Octβ2R* had an impact on the ability of TA to promote longevity (**Figure 7F**). In *TIGS>Octβ2R^{RNAi}* females, TA had a significant effect on lifespan in the absence of the RU486 inducer ($p=0.002$, *log-rank* test) while its effect on survival was not significant in the presence of RU486 ($p=0.8$, *log-rank* test). Indeed, CPH analysis confirmed that the response to TA was modulated by *Octβ2R* knockdown (RU486-by-TA feeding interaction $p=0.03$, **Supplementary Figure 9D**). Note that the effect of *Octβ2R^{RNAi}* observed is likely limited by the incomplete knock down of *Octb2R* mRNA (**Supplementary Figure 9E**, also reported in ref.⁶⁹)

We believe the results show that the effects of TA on lifespan are diminished upon knockdown of *Octβ2R* and are consistent with our conclusion that TA, *Octβ2R* and *CrebB* map onto the same longevity pathway. We hope that their inclusion in the new revision satisfies the request of the reviewer.

Reviewer #3 (Remarks to the Author):

The authors have addressed most of my concerns, and the manuscript is significantly improved. However, point 3 was inadequately addressed.

Throughout the manuscript, the authors rely on global genetic overexpression strategies (*Tdc2-Gal4 > UAS-Tdc2* in Figures 1-2; *Tbh-Gal4 > UAS-Tbh* in Figure 3) and systemic feeding experiments (Figure 4) to manipulate tyramine (TA) levels. However, these approaches fail to resolve fundamental questions regarding TA's cellular origin, spatiotemporal regulation during aging, and the basis for observed sexual dimorphism.

Critically, the source(s) of physiologically relevant TA pools remain unidentified. Potential contributors—including specific neuronal subtypes (CNS or enteric), enteroendocrine cells, or gut microbiota—were not distinguished experimentally. To

establish mechanistic causality, cell-type-specific manipulation (e.g., intersectional genetics) and temporal control of TA synthesis are essential. Furthermore, the work does not address how TA production is dynamically regulated throughout aging or how sex-specific differences arise.

The current model lacks spatial and temporal resolution, need more experimental evidence in addition to more discussion.

Previous point 3 of this reviewer:

3, The author is likely to present a model that the neuronal TA promotes gut PKA/CREB activity through the brain-gut axis. If it is the case, the authors should manipulate TA level in the brain while check the enteric PKA/CREB activity simultaneously. LexA combined with Gal4 could be used to address this question.

Our previous reply:

We are not presenting a model including a brain-gut axis. The key conclusion of our manuscript is that the activation of β -adrenergic-like signalling can promote longevity. Our data are consistent with either neuronal, endocrine/paracrine or microbial source of TA that acts on the signalling pathway within the gut. We have now made this clear in the discussion.

We thank the reviewer for appreciating the improvements we have made to the manuscript. The reviewer states that our response to point 3 is still inadequate. This may be because we did not explicitly comment on the suggested use of LexA combined with Gal4. We apologise for this. Essentially, to ensure relevance and robustness of our findings we work in an outbred population with all constructs backcrossed and backcrossing refreshed frequently to avoid genetic drift, selection of suppressors or bottlenecks. This makes it very difficult to work with flies that carry multiple transgenes. Colleagues at the institute have invested quite some time in trying to combine LexA and Gal4/GeneSwitch for ageing studies, however this has repeatedly proven problematic. Furthermore, the repertoire of LexA reagents is very, very limited. Essentially, combining LexA and Gal4 induction systems while

maintaining the experimental conditions that ensure relevance and robustness for ageing studies is currently not possible.

Additionally, we have to politely disagree with the reviewer's additional comments: while the biological questions raised by the reviewer are very interesting, such as the question of neuronal subtypes, they are completely beyond the scope of the current manuscript. We have very clearly argued this in the previous reply to the reviewer's other comments and included substantial changes in the text of the first revision to account for the reviewer's point of view.

Our key conclusion, that boosting β -adrenergic signalling in a specific organ can promote longevity, does not depend on addressing any of the questions raised by the reviewer. Indeed, this conclusion is valid irrespective of the source of TA, and supported by experiments that do not involve manipulation of TA levels. We think our findings are very relevant and interesting and need to be communicated without unnecessary delays. Some of the questions raised by the reviewer will indeed be followed up in subsequent studies.

References

Sujkowski, A., Gretzinger, A., Soave, N., Todi, S.V., and Wessells, R. (2020). Alpha- and beta-adrenergic octopamine receptors in muscle and heart are required for *Drosophila* exercise adaptations. *PLoS genetics* 16, e1008778.